# Anchor-based Maximum Discrepancy for Relative Similarity Testing

**Zhijian Zhou, Liuhua Peng, Xunye Tian, Feng Liu**[*]
The University of Melbourne, Australia
{zhijianzhou.ml, xunyetian.ml, fengliu.ml}@gmail.com
liuhua.peng@unimelb.edu.au

## Abstract

The *relative similarity testing* aims to determine which of the distributions, $\mathbb{P}$ or $\mathbb{Q}$, is closer to an anchor distribution $\mathbb{U}$. Existing kernel-based approaches often test the relative similarity with *a fixed kernel* in a *manually specified* alternative hypothesis, e.g., $\mathbb{Q}$ is closer to $\mathbb{U}$ than $\mathbb{P}$. Although kernel selection is known to be important to kernel-based testing methods, the manually specified hypothesis poses a significant challenge for kernel selection in relative similarity testing: Once the hypothesis is specified *first*, we can always find a kernel such that the hypothesis is rejected. This challenge makes relative similarity testing ill-defined when we want to select a good kernel *after* the hypothesis is specified. In this paper, we cope with this challenge via learning a proper hypothesis and a kernel *simultaneously*, instead of learning a kernel after manually specifying the hypothesis. We propose an *anchor-based maximum discrepancy* (AMD), which defines the relative similarity as the maximum discrepancy between the distances of $(\mathbb{U}, \mathbb{P})$ and $(\mathbb{U}, \mathbb{Q})$ in a space of deep kernels. Based on AMD, our testing incorporates two phases. In Phase I, we estimate the AMD over the deep kernel space and infer the *potential hypothesis*. In Phase II, we assess the statistical significance of the potential hypothesis, where we propose a unified testing framework to derive thresholds for tests over different possible hypotheses from Phase I. Lastly, we validate our method theoretically and demonstrate its effectiveness via extensive experiments on benchmark datasets. Codes are publicly available at: https://github.com/tmlr-group/AMD.

## 1 Introduction

Assessing differences between distributions is an important research area in machine learning, especially for tasks that often involve various distributions, e.g., different training and test data distributions in many real-world scenarios [1]. For such tasks, two-sample testing has been developed as a principled method to assess whether two distributions are identical [2]. However, it becomes less effective when the task requires comparing three or more distributions. For example, in scenarios where multiple models are available for a given test dataset, selecting the model trained on data most similar to the test data is essential for optimal performance. Additionally, in studies with generative data, including those arising from generative models [3], adversarial attacks [4], data augmentation [5] and machine generated text detection [6], a key challenge is to assess relative similarity between different generative data distributions and the original data distribution.

Fortunately, *relative similarity testing* provides a framework to assess which of the distributions, $\mathbb{P}$ or $\mathbb{Q}$, is closer to the anchor distribution $\mathbb{U}$; in practice, these distributions are typically unknown, and we can only observe samples from them. Existing methods evaluate the relative similarity between the pairs $(\mathbb{U}, \mathbb{P})$ and $(\mathbb{U}, \mathbb{Q})$ by comparing the discrepancy between their distances, $d(\mathbb{U}, \mathbb{P})$ and $d(\mathbb{U}, \mathbb{Q})$.

---

[*]Correspondence to Feng Liu (fengliu.ml@gmail.com).

39th Conference on Neural Information Processing Systems (NeurIPS 2025).

A popular distance is the *integral probability metric* (IPM) [7, 8], which defines the distance between two distributions as the maximum discrepancy between their expectations over a specified class of functions, offering strong theoretical guarantees and adaptability across various scenarios through the choice of function classes. The kernel-based *maximum mean discrepancy* (MMD) is an instance of IPM [9, 10], providing an effective measure for assessing distributional discrepancy [2, 11, 12], and is well-suited for applications in relative similarity testing [13, 14]. Some methods use kernelized Stein discrepancy, another instance of IPM, to evaluate similarity in cases where $\mathbb{U}$ is known [15–18].

Prior to conducting hypothesis testing, it is typically necessary to formulate the null and alternative hypotheses. Previous methods for relative similarity testing follow this procedure by *manually specifying a relationship* in the alternative hypothesis and performing the test accordingly. For example, given the specified relationship $d(\mathbb{U}, \mathbb{P}) > d(\mathbb{U}, \mathbb{Q})$, they define the hypotheses as follows

$$\boldsymbol{H}_0' : d(\mathbb{U}, \mathbb{P}) \leq d(\mathbb{U}, \mathbb{Q}) \text{ and } \boldsymbol{H}_1' : d(\mathbb{U}, \mathbb{P}) > d(\mathbb{U}, \mathbb{Q});$$

and then the difference between distances, i.e., $d(\mathbb{U}, \mathbb{P}) - d(\mathbb{U}, \mathbb{Q})$, is estimated and compared against a positive threshold to test the prespecified relative similarity relationship [13]. Consequently, instead of testing which of $\mathbb{P}$ or $\mathbb{Q}$ is closer to $\mathbb{U}$, these methods evaluate whether $\mathbb{Q}$ is closer to $\mathbb{U}$ than $\mathbb{P}$.

**Influence of Manually Specified Hypothesis in Oriented Test.** This type of oriented test focuses on evidence supporting $d(\mathbb{U}, \mathbb{P}) > d(\mathbb{U}, \mathbb{Q})$, and potentially neglects cases where $d(\mathbb{U}, \mathbb{P}) \leq d(\mathbb{U}, \mathbb{Q})$. We consider a scenario where $d(\mathbb{U}, \mathbb{P}) \leq d(\mathbb{U}, \mathbb{Q})$ and analyze the *probability $\epsilon$ of correctly identifying this relative similarity relationship*. It is evident that $\epsilon$ is equal to $1 - \alpha$ in the oracle hypothesis setting, which corresponds to the probability of accepting the null hypothesis $\boldsymbol{H}_0' : d(\mathbb{U}, \mathbb{P}) \leq d(\mathbb{U}, \mathbb{Q})$ given the significance level $\alpha$. This implies that there is a probability of $\alpha$ of neglecting the case $d(\mathbb{U}, \mathbb{P}) \leq d(\mathbb{U}, \mathbb{Q})$, regardless of the amount of data or the kernel used for testing. Thus, it is crucial to construct an alternative hypothesis that reflects the potential true relative similarity relationship, which is unknown in practice. Without prior knowledge, randomly selecting an alternative hypothesis results in a success rate of merely $0.5$.

**Kernel Selection Challenge for Existing Relative Similarity Tests.** More importantly, in kernel-based test, it is critical to select a kernel to effectively capture the evidence that supports the alternative hypothesis [19], and the relative similarity relationship may vary with the choice of kernel from the same family, such as Gaussian kernels. For instance, with a Gaussian kernel $\kappa_A$, the relationship $d(\mathbb{U}, \mathbb{P}) > d(\mathbb{U}, \mathbb{Q})$ may hold, while with Gaussian kernel $\kappa_B$, $d(\mathbb{U}, \mathbb{P}) < d(\mathbb{U}, \mathbb{Q})$ holds instead. Given this, *after* we specify a relative similarity relationship in alternative hypothesis as in $\boldsymbol{H}_1'$, the method then tends to select a kernel that best supports $d(\mathbb{U}, \mathbb{P}) > d(\mathbb{U}, \mathbb{Q})$. This deviates from the primary goal of testing which distribution, $\mathbb{P}$ or $\mathbb{Q}$, is closer to the anchor distribution $\mathbb{U}$. A median heuristics selects the Gaussian kernel bandwidth by averaging median distances between samples from $\mathbb{U}$ and $\mathbb{P}$, and those from $\mathbb{U}$ and $\mathbb{Q}$ [13], but it has no guarantees of effective performance in various scenarios [20, 19] or with complex data (e.g., images) [21, 22]. Moreover, it restricts the use of more expressive kernels, such as deep kernel [21] and Mahalanobis kernel [23].

**Our Solution.** In this paper, we cope with the above challenges via learning a proper hypothesis and a kernel *simultaneously*, instead of learning a kernel after manually specifying the hypothesis. We propose an *anchor-based maximum discrepancy* (AMD), which defines the relative similarity as the maximum discrepancy between the distances of $(\mathbb{U}, \mathbb{P})$ and $(\mathbb{U}, \mathbb{Q})$ in a space of deep kernels (motivated by the definition of IPM). This approach inherently incorporates kernel selection and avoids a selection towards a prespecified relative similarity relationship, e.g., $d(\mathbb{U}, \mathbb{P}) > d(\mathbb{U}, \mathbb{Q})$. Based on the AMD metric, our relative similarity testing incorporates two phases

- In Phase I (Section 3.2), we estimate the AMD by selecting the kernel that maximizes the discrepancy between the distances of $(\mathbb{U}, \mathbb{P})$ and $(\mathbb{U}, \mathbb{Q})$ from a kernel space. Given this estimation, we infer the potential relative similarity relationship, that is, which distribution, $\mathbb{P}$ or $\mathbb{Q}$, is closer to $\mathbb{U}$, instead of manually specifying an interested relationship. Here, to ensure a sufficiently rich function space and enhance adaptability across various tasks, we use deep kernels built upon neural networks [21] and propose an optimization algorithm to estimate the AMD with augmented samples, which improves generalization and reduces overfitting for small sample sizes [24–26].

- In Phase II (Section 3.3), to assess the statistical significance of the relative similarity relationship identified in Phase 1, we perform the AMD test with the selected kernel from Phase I. In this procedure, we consider the tests on two possible relative similarity relationships, i.e., $\mathbb{P}$ is closer to $\mathbb{U}$ or $\mathbb{Q}$ is closer to $\mathbb{U}$. For this, we propose a unified testing framework to derive the testing thresholds for both cases using the wild bootstrap method.

Theoretically, our AMD serves as a valid metric, and its estimation with data augmentation is proven to exhibit consistency. Besides, we analyze the advantage of our AMD test in test power, while ensuring that the type-I error is bounded at the significance level. Empirically, we validate the AMD test against state-of-the-art methods on benchmark datasets, and further conduct ablation studies to evaluate the contribution of each component. We also illustrate practical applications of our AMD test in comparing model performance across different datasets relative to a reference dataset.

## 2   Preliminaries

**Integral Probability Metric.** Let $\mathbb{P}$ and $\mathbb{Q}$ denote two Borel probability measures over a space $\mathcal{X} \subseteq \mathbb{R}^d$. For a class $\mathcal{F}$ of functions, the *integral probability metric* (IPM) [7] is defined as the distance between the distributions $\mathbb{P}$ and $\mathbb{Q}$, as

$$d(\mathbb{P}, \mathbb{Q}; \mathcal{F}) = \sup_{f \in \mathcal{F}} \left| \int f \, d\mathbb{P} - \int f \, d\mathbb{Q} \right| .$$

This framework is particularly relevant in two-sample test, which aims to assess the equality between two unknown distributions. Numerous approaches have been proposed to find distance measures of this form, aimed at distinguishing between various distributions. Among them, a widely used approach is the kernel-based *Maximum Mean Discrepancy* (MMD), which corresponds to an IPM defined over the function class $\mathcal{H}_\kappa$, i.e., the Reproducing Kernel Hilbert space (RKHS) of a kernel $\kappa : \mathcal{X} \times \mathcal{X} \to \mathbb{R}$. The MMD serves as an effective distance for measuring the similarity between two distributions, satisfying $\mathrm{MMD}(\mathbb{P}, \mathbb{Q}; \kappa) = 0$ if and only if $\mathbb{P} = \mathbb{Q}$ for characteristic kernels [9].

**Relative Similarity Testing with MMD.** Using the squared MMD as similarity measure and given the anchor Borel probability measure $\mathbb{U}$ over $\mathcal{X} \subseteq \mathbb{R}^d$, Bounliphone et al. [13] test whether $\mathbb{Q}$ is closer to $\mathbb{U}$ than $\mathbb{P}$ by formulating the hypotheses as follows:

$$\boldsymbol{H}_0' : \mathrm{MMD}^2(\mathbb{U}, \mathbb{P}; \kappa) \le \mathrm{MMD}^2(\mathbb{U}, \mathbb{Q}; \kappa) \quad \text{and} \quad \boldsymbol{H}_1' : \mathrm{MMD}^2(\mathbb{U}, \mathbb{P}; \kappa) > \mathrm{MMD}^2(\mathbb{U}, \mathbb{Q}; \kappa) .$$

Given the alternative hypothesis, the test statistic is defined as $\mathrm{MMD}^2(\mathbb{U}, \mathbb{P}; \kappa) - \mathrm{MMD}^2(\mathbb{U}, \mathbb{Q}; \kappa)$, which can be equivalently expressed using the IPM as:

$$\left[ \sup_{f \in \mathcal{H}_\kappa} \left| \int f \, d\mathbb{U} - \int f \, d\mathbb{P} \right| \right]^2 - \left[ \sup_{g \in \mathcal{H}_\kappa} \left| \int g \, d\mathbb{U} - \int g \, d\mathbb{Q} \right| \right]^2 .$$

In practice, the three distributions $\mathbb{U}$, $\mathbb{P}$ and $\mathbb{Q}$ are generally unknown and what we can observe are three samples $Z = \{\boldsymbol{z}_i\}_{i=1}^m \sim \mathbb{U}^m$, $X = \{\boldsymbol{x}_i\}_{i=1}^m \sim \mathbb{P}^m$ and $Y = \{\boldsymbol{y}_i\}_{i=1}^m \sim \mathbb{Q}^m$[1]. By estimating the test statistic using samples, i.e., $\widehat{\mathrm{MMD}}^2(Z, X; \kappa) - \widehat{\mathrm{MMD}}^2(Z, Y; \kappa)$, and comparing it against a positive testing threshold corresponding to the given significance level $\alpha$, this approach effectively focuses on testing evidence in favor of the alternative hypothesis $\boldsymbol{H}_1'$. Furthermore, in kernel-based testing, it is critical to select a kernel to effectively capture the evidence that supports the alternative hypothesis, which leads to selecting a kernel favoured at the prespecified alternative hypothesis.

## 3   The AMD Relative Similarity Testing

As demonstrated in Section 1, although kernel selection is known to be important to kernel-based testing methods, the manually specified hypothesis poses a significant challenge for kernel selection in relative similarity testing: Once the hypothesis is specified *first*, we can always find a kernel such that the hypothesis is rejected. This challenge makes relative similarity testing ill-defined when we want to select a good kernel *after* the hypothesis is specified. In this section, we cope with the above challenges via learning a proper hypothesis and a kernel *simultaneously*, instead of learning a kernel after manually specifying the hypothesis.

We first introduce the relative similarity measure *anchor-based maximum discrepancy* (AMD) in Section 3.1; then in Section 3.2, we present the Phase I of AMD test: estimating AMD and inferring a potential relative similarity relationship, i.e., learning a proper hypothesis and a kernel *simultaneously*. Finally, in Phase II (Section 3.3), we test the statistical significance of the relationship from Phase I.

---

[1]As done in Liu et al. [21], we assume equal size for three samples to simplify the notation, yet the results in this paper can be easily extended to unequal sample sizes following approaches based on the $U$-statistic [27, 9].

## 3.1 The Anchor-based Maximum Discrepancy (AMD)

As introduced in Section 1, IPM is an effective metric in measuring the discrepancy between two distributions with strong theoretical guarantees and adaptability across various scenarios through the choice of function classes. Here, we extend it to the relative similarity testing with three distributions, which measures the maximum discrepancy between the distances of $(\mathbb{U}, \mathbb{P})$ and $(\mathbb{U}, \mathbb{Q})$. Specifically, we define the *anchor-based maximum discrepancy* (AMD) as follows

**Definition 1.** For each kernel $\kappa$ in the kernel family $\mathcal{K}$, we define function $f_\kappa = \langle \cdot, \boldsymbol{\mu}_{\mathbb{P}} - \boldsymbol{\mu}_{\mathbb{Q}} \rangle_{\mathcal{H}_\kappa}$ and introduce the corresponding function space $\mathcal{F}_\mathcal{K} = \{ f_\kappa | \kappa \in \mathcal{K} \}$. The AMD, i.e., $d(\mathbb{U}, \mathbb{P}, \mathbb{Q}; \mathcal{F}_\mathcal{K})$, is

$$\sup_{f_\kappa \in \mathcal{F}_\mathcal{K}} \left| \int f_\kappa \, d\mathbb{U} - \int f_\kappa \, d\frac{\mathbb{P} + \mathbb{Q}}{2} \right| = \sup_{\kappa \in \mathcal{K}} \left| \left\langle \boldsymbol{\mu}_{\mathbb{U}} - \frac{\boldsymbol{\mu}_{\mathbb{P}} + \boldsymbol{\mu}_{\mathbb{Q}}}{2}, \boldsymbol{\mu}_{\mathbb{P}} - \boldsymbol{\mu}_{\mathbb{Q}} \right\rangle_{\mathcal{H}_\kappa} \right|, \tag{1}$$

where $\boldsymbol{\mu}_{\mathbb{U}} = E_{\boldsymbol{z} \sim \mathbb{U}}[\kappa(\cdot, \boldsymbol{z})]$ and $\boldsymbol{\mu}_{\mathbb{P}}, \boldsymbol{\mu}_{\mathbb{Q}}$ are defined similarly with $\kappa(\cdot, \boldsymbol{x}) \in \mathcal{H}_\kappa$ as the feature map.

In AMD, we measure the relative similarity between distributions by their characteristic kernel mean embeddings (i.e., $\boldsymbol{\mu}_{\mathbb{U}}, \boldsymbol{\mu}_{\mathbb{P}}$, and $\boldsymbol{\mu}_{\mathbb{Q}}$). These embeddings uniquely represent probability distributions and capture their distinct characteristics for comparison [28]. The reason to choose such an $f_\kappa$ is that the Euclidean-like distances (i.e. MMD distances) $\|\boldsymbol{\mu}_{\mathbb{U}} - \boldsymbol{\mu}_{\mathbb{P}}\|_{\mathcal{H}_\kappa}^2$ and $\|\boldsymbol{\mu}_{\mathbb{U}} - \boldsymbol{\mu}_{\mathbb{Q}}\|_{\mathcal{H}_\kappa}^2$ are equal in RKHS $\mathcal{H}_\kappa$ if and only if $\boldsymbol{\mu}_{\mathbb{U}} - (\boldsymbol{\mu}_{\mathbb{P}} + \boldsymbol{\mu}_{\mathbb{Q}})/2$ is orthogonal to vector $\boldsymbol{\mu}_{\mathbb{P}} - \boldsymbol{\mu}_{\mathbb{Q}}$. Here, AMD measures relative similarity as the maximum discrepancy within the kernel space $\mathcal{K}$, avoiding oriented kernel selection arising from prespecified relative similarity relationship, as discussed in Section 1.

We now prove that our AMD (i.e., Eqn. (1)) is a valid metric as follows.

**Theorem 2.** *Let $\mathcal{M}$ be a set of probability measures over the space $\mathcal{X} \subseteq \mathbb{R}^d$, and let $\mathcal{K}$ be a kernel space consisting of characteristic kernels. For every $\mathbb{U}, \mathbb{P}, \mathbb{Q}, \mathbb{W} \in \mathcal{M}$, the three random variables $d(\mathbb{U}, \mathbb{P}, \mathbb{Q}; \mathcal{F}_\mathcal{K})$, $d(\mathbb{U}, \mathbb{P}, \mathbb{W}; \mathcal{F}_\mathcal{K})$ and $d(\mathbb{U}, \mathbb{Q}, \mathbb{W}; \mathcal{F}_\mathcal{K})$ satisfy:*

- $d(\mathbb{U}, \mathbb{P}, \mathbb{Q}; \mathcal{F}_\mathcal{K}) \geq 0$;
- $d(\mathbb{U}, \mathbb{P}, \mathbb{Q}; \mathcal{F}_\mathcal{K}) = d(\mathbb{U}, \mathbb{Q}, \mathbb{P}; \mathcal{F}_\mathcal{K})$;
- $d(\mathbb{U}, \mathbb{P}, \mathbb{Q}; \mathcal{F}_\mathcal{K}) \leq d(\mathbb{U}, \mathbb{P}, \mathbb{W}; \mathcal{F}_\mathcal{K}) + d(\mathbb{U}, \mathbb{Q}, \mathbb{W}; \mathcal{F}_\mathcal{K})$.

## 3.2 Phase 1: Estimating AMD and Infer the Potential Relative Similarity Relationship

Based on Eqn. (1), AMD can be expressed as $d(\mathbb{U}, \mathbb{P}, \mathbb{Q}; \mathcal{F}_\mathcal{K}) = \sup_{\kappa \in \mathcal{K}} |d^\kappa(\mathbb{U}, \mathbb{P}, \mathbb{Q})|$ with

$$d^\kappa(\mathbb{U}, \mathbb{P}, \mathbb{Q}) = \left\langle \boldsymbol{\mu}_{\mathbb{U}} - \frac{\boldsymbol{\mu}_{\mathbb{P}} + \boldsymbol{\mu}_{\mathbb{Q}}}{2}, \boldsymbol{\mu}_{\mathbb{P}} - \boldsymbol{\mu}_{\mathbb{Q}} \right\rangle_{\mathcal{H}_\kappa} = E\left[ \kappa(\boldsymbol{z}, \boldsymbol{x}) - \kappa(\boldsymbol{z}, \boldsymbol{y}) - \frac{\kappa(\boldsymbol{x}, \boldsymbol{x}')}{2} + \frac{\kappa(\boldsymbol{y}, \boldsymbol{y}')}{2} \right],$$

where $\boldsymbol{z} \sim \mathbb{U}$, $\boldsymbol{x}, \boldsymbol{x}' \sim \mathbb{P}^2$ and $\boldsymbol{y}, \boldsymbol{y}' \sim \mathbb{Q}^2$. Here, by the geometric relationship between the vectors $\boldsymbol{\mu}_{\mathbb{U}} - (\boldsymbol{\mu}_{\mathbb{P}} + \boldsymbol{\mu}_{\mathbb{Q}})/2$ and $\boldsymbol{\mu}_{\mathbb{P}} - \boldsymbol{\mu}_{\mathbb{Q}}$, we interpret the sign of $d^\kappa(\mathbb{U}, \mathbb{P}, \mathbb{Q})$ as an indicator of relative similarity relationship: $\mathbf{sgn}(d^\kappa(\mathbb{U}, \mathbb{P}, \mathbb{Q})) = 1$ **indicates that $\mathbb{P}$ is closer to $\mathbb{U}$; while** $\mathbf{sgn}(d^\kappa(\mathbb{U}, \mathbb{P}, \mathbb{Q})) = -1$ **indicates that $\mathbb{Q}$ is closer to $\mathbb{U}$.**

For unknown distributions $\mathbb{U}$, $\mathbb{P}$ and $\mathbb{Q}$ with corresponding samples $Z = \{\boldsymbol{z}_i\}_{i=1}^m \sim \mathbb{U}^m$, $X = \{\boldsymbol{x}_i\}_{i=1}^m \sim \mathbb{P}^m$ and $Y = \{\boldsymbol{y}_i\}_{i=1}^m \sim \mathbb{Q}^m$, we estimate the measure $d^\kappa(\mathbb{U}, \mathbb{P}, \mathbb{Q})$ use the *U-statistc* estimator, known to be the unbiased estimator with minimum variance [27], as follows

$$\widehat{d^\kappa}(Z, X, Y) = \sum_{i \neq j} \frac{\kappa(\boldsymbol{z}_i, \boldsymbol{x}_j) + \kappa(\boldsymbol{z}_j, \boldsymbol{x}_i) - \kappa(\boldsymbol{z}_i, \boldsymbol{y}_j) - \kappa(\boldsymbol{z}_j, \boldsymbol{y}_i) - \kappa(\boldsymbol{x}_i, \boldsymbol{x}_j) + \kappa(\boldsymbol{y}_i, \boldsymbol{y}_j)}{2m(m-1)} . \tag{2}$$

Estimating the AMD is equivalent to selecting the optimal kernel that maximizes the statistic $\widehat{d^\kappa}(Z, X, Y)$, which can be formulated as the following optimization objective

$$\kappa_m^* \in \arg\max_{\kappa \in \mathcal{K}} |\widehat{d^\kappa}(Z, X, Y)| . \tag{3}$$

The absolute value introduces challenges to optimization. A common strategy to address this is to optimize the squared form, i.e. $\widehat{d^\kappa}(Z, X, Y)^2$, which would enhances differentiability and facilitates a smoother optimization [29]. However, this method has a drawback: it is highly sensitive to outliers or

data points with large values [30], especially in high-density regions. This can overshadow important patterns in lower-density regions, potentially leading to an incomplete understanding of data.

To mitigate this issue, the optimization is divided into two separate cases: 1) $\widehat{d^\kappa}(Z, X, Y) > 0$; 2) $\widehat{d^\kappa}(Z, X, Y) < 0$. For each case, the optimal kernel is selected independently, and the final kernel is chosen as the one that achieves the highest absolute value of the metric. Treating these cases individually helps to better capture the characteristics of the data in both high- and low-density regions. Yet, this approach may still result in sub-optimal solutions. This is because directly maximizing or minimizing the estimator can amplify specific data patterns excessively, increasing the risk of overfitting [], particularly when sample size is limited and when a deep kernel[2] is employed as in [21]

$$\kappa(\boldsymbol{x}, \boldsymbol{y}) = [(1 - \epsilon)G_1(\phi_\omega(\boldsymbol{x}), \phi_\omega(\boldsymbol{y})) + \epsilon]G_2(\boldsymbol{x}, \boldsymbol{y}) , \tag{4}$$

with Gaussian kernels $G_1$ and $G_2$, and a neural network $\phi_\omega$.

**Augmentation-assisted Estimation of AMD.** We mitigate the overfitting issue via an iterative optimization process known as data augmentation [5]. However, the distribution of augmented samples normally deviates from that of the original samples, which leads to an inconsistent estimation of the AMD metric. Here, we construct augmented samples $Z^{\mathrm{aug}}$, $X^{\mathrm{aug}}$ and $Y^{\mathrm{aug}}$ under the condition that $E[\widehat{d^\kappa}(Z^{\mathrm{aug}}, X^{\mathrm{aug}}, Y^{\mathrm{aug}})] = 0$ to ensure the consistency. Specifically, at each iteration, with $\mathbb{U}$ as the anchor distribution, we generate $Z^{\mathrm{aug}} = \{\boldsymbol{z}_i^{\mathrm{aug}}\}_{i=1}^m \sim Z^m$, which is an i.i.d copy of $Z$ and maintains statistical properties of $\mathbb{U}$. Then, for $X^{\mathrm{aug}} = \{\boldsymbol{x}_i^{\mathrm{aug}}\}_{i=1}^m$ and $Y^{\mathrm{aug}} = \{\boldsymbol{y}_i^{\mathrm{aug}}\}_{i=1}^m$, we have

$$\boldsymbol{x}_i^{\mathrm{aug}} = 0.5\boldsymbol{x} + 0.5\boldsymbol{y} \qquad \text{and} \qquad \boldsymbol{y}_i^{\mathrm{aug}} = 0.5\boldsymbol{x}' + 0.5\boldsymbol{y}' \qquad \text{with} \qquad \boldsymbol{x}, \boldsymbol{x}' \sim X, \ \boldsymbol{y}, \boldsymbol{y}' \sim Y . \tag{5}$$

It is evident that two samples $X^{\mathrm{aug}}$ and $Y^{\mathrm{aug}}$ are drawn from the identical distribution, and are thus equally close to the anchor distribution $\mathbb{U}$ satisfying the condition $E[\widehat{d^\kappa}(Z^{\mathrm{aug}}, X^{\mathrm{aug}}, Y^{\mathrm{aug}})] = 0$. We take $\widehat{d^\kappa}(Z^{\mathrm{aug}}, X^{\mathrm{aug}}, Y^{\mathrm{aug}})$ as a regularization term to avoid overfitting and focus the optimization on $\widehat{d^\kappa}(Z, X, Y)$, which is the main objective for estimating the AMD. For case $\widehat{d^\kappa}(Z, X, Y) > 0$, the optimization with a regularization parameter $0 \le \lambda < \infty$ is

$$\kappa_m^{*,+} \in \arg\max_{\kappa \in \mathcal{K}} \widehat{d^\kappa}(Z, X, Y) - \lambda \cdot \widehat{d^\kappa}(Z^{\mathrm{aug}}, X^{\mathrm{aug}}, Y^{\mathrm{aug}})^2 ;$$

in a similar manner, for $\widehat{d^\kappa}(Z, X, Y) < 0$, we perform optimization as

$$\kappa_m^{*,-} \in \arg\min_{\kappa \in \mathcal{K}} \widehat{d^\kappa}(Z, X, Y) + \lambda \cdot \widehat{d^\kappa}(Z^{\mathrm{aug}}, X^{\mathrm{aug}}, Y^{\mathrm{aug}})^2 .$$

With an expected value of zero, the statistic $\widehat{d^\kappa}(Z^{\mathrm{aug}}, X^{\mathrm{aug}}, Y^{\mathrm{aug}})$ actually reflects sampling variability. We square it as regularization term to account for reduce the impact of fluctuations where the empirical estimator differs from its expected value, thereby mitigating the overfitting in the estimation.

Finally, we select the optimal kernel as follows

$$\kappa_m^* = \arg\max_{\kappa \in \{\kappa_m^{*,+}, \kappa_m^{*,-}\}} |\widehat{d^\kappa}(Z, X, Y)| . \tag{6}$$

**Infer the potential relative similarity relationship.** Unlike previous approaches that manually specify a relative similarity relationship in alternative hypothesis, our method allows for inferring the potential relationship with kernel $\kappa_m^*$. The inferred relationship is denoted by $F$, where

- $F = 1$ if $\kappa_m^* = \kappa_m^{*,+}$, indicating that distribution $\mathbb{P}$ is closer to the anchor distribution $\mathbb{U}$.
- $F = -1$ if $\kappa_m^* = \kappa_m^{*,-}$, indicating that distribution $\mathbb{Q}$ is closer to the anchor distribution $\mathbb{U}$.

**Consistency of the Estimation.** We now establish the consistency of estimating $d(\mathbb{U}, \mathbb{P}, \mathbb{Q}; \mathcal{F_K})$ via the optimization procedure with augmented samples, under Assumption 1 (Appendix A.2).

**Theorem 3.** *Let $\mathcal{K}$ be a kernel space consisting of symmetric kernels $\kappa$ satisfying $0 \le \kappa(\boldsymbol{x}, \boldsymbol{y}) \le B$ for all $\boldsymbol{x}, \boldsymbol{y} \in \mathcal{X}$. Then, as $m \to \infty$ and $0 \le \lambda < \infty$, the following holds*

$$F \cdot d^{\kappa_m^*}(Z, X, Y) - d(\mathbb{U}, \mathbb{P}, \mathbb{Q}; \mathcal{F_K}) \xrightarrow{\text{a.s.}} 0 .$$

This theorem shows that, although we perform estimation with augmented data, the estimator $F \cdot d^{\kappa_m^*}(Z, X, Y)$ almost surely converges to the true AMD value as sample size approaches infinity.

---

[2]Our methods are also applicable to Laplace [12], Mahalanobis [23] and Gaussian kernels [31], etc, which can be optimizaed with

### 3.3 Phase II: Testing the Statistical Significance of Potential Relative Similarity Relationship

Next, we assess the statistical significance of the relative similarity relationship identified in Phase I. In the testing procedure, we use testing samples $Z' = \{z_i'\}_{i=1}^m \sim \mathbb{U}^m$, $X' = \{x_i'\}_{i=1}^m \sim \mathbb{P}^m$ and $Y' = \{y_i'\}_{i=1}^m \sim \mathbb{Q}^m$, which are drawn independently of $Z$, $X$, and $Y$. This follows the data-splitting strategy commonly used in kernel-based hypothesis testing to ensure the validity of the test [21].

Given $F$ from Phase 1, we consider tests for both relative similarity relationships with $d^{\kappa_m^*}(\mathbb{U}, \mathbb{P}, \mathbb{Q})$ (i.e., a single kernel form of AMD with the consistently estimated kernel $\kappa_m^*$), which is defined as

- For $F = 1$, indicating a potential relationship that $\mathbb{P}$ is closer to $\mathbb{U}$, we test on hypotheses

$$\boldsymbol{H}_0 : d^{\kappa_m^*}(\mathbb{U}, \mathbb{P}, \mathbb{Q}) \leq 0 \qquad \text{and} \qquad \boldsymbol{H}_1 : d^{\kappa_m^*}(\mathbb{U}, \mathbb{P}, \mathbb{Q}) > 0 \,.$$

  Here, given the testing threshold $\tau_\alpha > 0$ for a significance level $\alpha$, we conclude that $\mathbb{P}$ is significantly closer to $\mathbb{U}$ than $\mathbb{P}$ at level $\alpha$ if test statistic satisfies $\widehat{d}^{\kappa_m^*}(Z', X', Y') > \tau_\alpha$.

- For $F = -1$, indicating a potential relationship that $\mathbb{Q}$ is closer to $\mathbb{U}$, we test on hypotheses

$$\boldsymbol{H}_0 : d^{\kappa_m^*}(\mathbb{U}, \mathbb{P}, \mathbb{Q}) \geq 0 \qquad \text{and} \qquad \boldsymbol{H}_1 : d^{\kappa_m^*}(\mathbb{U}, \mathbb{P}, \mathbb{Q}) < 0 \,.$$

  In a similar manner, given the testing threshold $\tau_\alpha < 0$ for a significance level $\alpha$, we conclude that $\mathbb{Q}$ is significantly closer to $\mathbb{U}$ than $\mathbb{Q}$ at level $\alpha$ if test statistic satisfies $\widehat{d}^{\kappa_m^*}(Z', X', Y') < \tau_\alpha$.

Then, based on the value of $F$, we can unify the above hypotheses as one:

$$\boldsymbol{H}_0^{\mathrm{uni}} : F \cdot d^{\kappa_m^*}(\mathbb{U}, \mathbb{P}, \mathbb{Q}) \leq 0 \qquad \text{and} \qquad \boldsymbol{H}_1^{\mathrm{uni}} : F \cdot d^{\kappa_m^*}(\mathbb{U}, \mathbb{P}, \mathbb{Q}) > 0 \,.$$

Given the testing threshold $F \cdot \tau_\alpha > 0$, the testing procedures can be formalized as

$$\mathfrak{h}(Z', X', Y'; \kappa_m^*) = \mathbb{I}[F \cdot \widehat{d}^{\kappa_m^*}(Z', X', Y') > F \cdot \tau_\alpha] \,, \tag{7}$$

where $\mathfrak{h}(Z', X', Y'; \kappa_m^*) = 1$ indicates that the inferred relative similarity relationship presented by $F$ is statistically significant at level $\alpha$; otherwise, the relationship is not statistically significant.

The critical step here is to determine the testing threshold $F \cdot \tau_\alpha$, which bounds the type-I error under the null hypothesis, i.e., $\Pr\left(\mathfrak{h}(Z', X', Y'; \kappa_m^*) = 1\right) \leq \alpha$. Notably, the unified null hypothesis $\boldsymbol{H}_0^{\mathrm{uni}}$ is composite, consisting of the case $F \cdot d^{\kappa_m^*}(\mathbb{U}, \mathbb{P}, \mathbb{Q}) = 0$ and the case where $F \cdot d^{\kappa_m^*}(\mathbb{U}, \mathbb{P}, \mathbb{Q}) < 0$. Since the ground-truth $d^{\kappa_m^*}(\mathbb{U}, \mathbb{P}, \mathbb{Q})$ is unknown, we set testing threshold $F \cdot \tau_\alpha$ as the $(1 - \alpha)$-quantile of the estimated distribution of $F \cdot \widehat{d}^{\kappa_m^*}(Z', X', Y')$ under *the proxy null hypothesis $\boldsymbol{H}_0^{\mathrm{p}}$*: $F \cdot d^{\kappa_m^*}(\mathbb{U}, \mathbb{P}, \mathbb{Q}) = 0$ (i.e., the least-favorable boundary of the composite null hypothesis) by wild bootstrap. Specifically, let $B$ be the number of bootstraps. In the $b$-th iteration ($b \in [B]$), we draw i.i.d. variables $\boldsymbol{\xi} = (\xi_1, ..., \xi_m)$ from Exponential distribution with scale parameter 1 and define

$$\boldsymbol{\zeta} = \{\zeta_i\}_{i=1}^m \quad \text{with} \quad \zeta_i = m \cdot \xi_i / \sum_j^m \xi_j \,.$$

Then, we calculate the $b$-th wild bootstrap statistic for $\widehat{d}^{\kappa_m^*}(Z', X', Y')$ as follows

$$T_b = \sum_{i \neq j} (\zeta_i \zeta_j - 1) \times \frac{\kappa(z_i', x_j') + \kappa(z_j', x_i') - \kappa(z_i', y_j') - \kappa(z_j', y_i') - \kappa(x_i', x_j') + \kappa(y_i', y_j')}{2m(m-1)} \,.$$

During such process, we obtain $B$ statistics $T_1, T_2, ..., T_B$ and introduce testing threshold $F \cdot \tau_\alpha$ with

$$\tau_\alpha = \arg\min_\tau \left\{ \sum_{b=1}^B \frac{\mathbb{I}[F \cdot T_b \leq F \cdot \tau]}{B} \geq 1 - \alpha \right\} \,. \tag{8}$$

We present theoretical analysis for type-I error as follows.

**Lemma 4.** *The type-I error of the unified testing procedure is bounded by $\alpha$.*

**Output of our AMD relative similarity test.** The output consist of two components: $F$ from Phase 1 and $\mathfrak{h}(Z', X', Y'; \kappa_m^*)$ from Phase 2. The term $F$ specifies the nature of the potential relative similarity relationship: 1) $F = 1$ indicates $\mathbb{P}$ is closer to $\mathbb{U}$; 2) $F = -1$ indicates $\mathbb{Q}$ is closer to $\mathbb{U}$. The term $\mathfrak{h}(Z', X', Y'; \kappa_m^*)$ determines whether the inferred relationship $F$ is significant at level $\alpha$.

We present the details of our AMD relative similarity testing in Algorithm 1. To help situate our methodology and highlight key differences, we introduce relevant works in Appendix B.

**Algorithm 1** The AMD test

---

**Input**: Training Samples $Z$, $X$ and $Y$, Iteration Epochs $T$ for training, Testing Samples $Z'$, $X'$ and $Y'$, Iteration Epochs $B$ for testing
**Initialization**: Select $\kappa_m^{*,+} = \kappa_m^{*,-}$ according to the median heuristic [13].
# *Phase 1: Select Kernel $\kappa_m^*$ and relationship $F$*

   **for** $t = 1, \ldots, T$ **do**
      Randomly generate augmented samples $Z^{\mathrm{aug}}$, $X^{\mathrm{aug}}$ and $Y^{\mathrm{aug}}$ as specified in Eqn. (5)

$$\kappa_m^{*,+} \;\leftarrow\; \kappa_m^{*,+} + \eta \cdot \nabla \left( \widehat{d}^{\kappa_m^{*,+}}(Z, X, Y) - \lambda \cdot \widehat{d}^{\kappa_m^{*,+}}(Z^{\mathrm{aug}}, X^{\mathrm{aug}}, Y^{\mathrm{aug}})^2 \right)$$

$$\kappa_m^{*,-} \;\leftarrow\; \kappa_m^{*,-} - \eta \cdot \nabla \left( \widehat{d}^{\kappa_m^{*,-}}(Z, X, Y) + \lambda \cdot \widehat{d}^{\kappa_m^{*,-}}(Z^{\mathrm{aug}}, X^{\mathrm{aug}}, Y^{\mathrm{aug}})^2 \right)$$

   **end for**
   Select $\kappa_m^*$ as in Eqn. (6) and relative similarity relationship $F$
# *Phase 2: testing with kernel $\kappa_m^*$ and relationship $F$*

   **for** $b = 1, \ldots, B$ **do**
      Draw $\boldsymbol{\xi} = (\xi_1, \xi_2, ..., \xi_m)$ from distribution $\mathrm{Exp}(1)$
      Let $\boldsymbol{\zeta} = \{\zeta_i\}_{i=1}^m$    with    $\zeta_i = m \cdot \xi_i / \sum_j^m \xi_j$

$$T_b \leftarrow \frac{1}{m(m-1)} \sum_{i \neq j} (\zeta_i \zeta_j - 1) h_{i,j}^{\kappa_m^*}(Z', X', Y')/2$$

   **end for**

$$\tau_\alpha \leftarrow \arg\min_\tau \left\{ \sum_{b=1}^B \frac{\mathbb{I}[F \cdot T_b \leq F \cdot \tau]}{B} \geq 1 - \alpha \right\}$$

$$\mathfrak{h}(Z', X', Y'; \kappa_m^*) = \mathbb{I}[F \cdot \widehat{d}^{\kappa_m^*}(Z', X', Y') > \tau_\alpha]$$

**Output**: $\mathfrak{h}(Z', X', Y'; \kappa_m^*)$ and $F$.

---

## 4   Asymptotic Test Power Comparison of Testing Procedures

Next, to analyze the advantages of our method in test power, we consider a practical scenario in which a relative similarity relationship is present, indicated by $d^{\kappa_m^*}(\mathbb{U}, \mathbb{P}, \mathbb{Q}) \neq 0$. However, the specific nature of this relationship, represented by $\mathrm{sgn}(d^{\kappa_m^*}(\mathbb{U}, \mathbb{P}, \mathbb{Q})) \in \{-1, 1\}$, remains unknown. Building on this, we denote the probability that $F$ captures the true relative similarity relationship as

$$\beta = \Pr[F \cdot d^{\kappa_m^*}(\mathbb{U}, \mathbb{P}, \mathbb{Q}) > 0] \,. \tag{9}$$

**Comparison with the test on a uniformly selected alternative hypothesis.** Previous methods [13, 14] require manually specifying a relative similarity relationship in alternative hypothesis (e.g., $d^{\kappa_m^*}(\mathbb{U}, \mathbb{P}, \mathbb{Q}) > 0$) and perform test accordingly. Yet, the prespecified relationship may not align with the true relationship. In the practical scenario, if we uniformly select an alternative hypothesis, the probability that it reflects the true relative similarity relationship is 0.5.

Now, we present the comparison of test power with a uniformly selected alternative hypothesis as

**Theorem 5.** *If $d^{\kappa_m^*}(\mathbb{U}, \mathbb{P}, \mathbb{Q}) \neq 0$, the test based on the learned alternative hypothesis with $F$ achieves a higher test power than that based on a uniformly selected alternative hypothesis as follows*

$$(\beta - 0.5) \Phi \left( \sqrt{m} \frac{|d^{\kappa_m^*}(\mathbb{U}, \mathbb{P}, \mathbb{Q})| - F \cdot \tau_\alpha}{\sigma_{\mathbb{U}, \mathbb{P}, \mathbb{Q}}} \right) + O(e^{-m}) \,,$$

*where where $\Phi(\cdot)$ is the cumulative distribution function of the standard normal distribution and $\sigma_{\mathbb{U}, \mathbb{P}, \mathbb{Q}}$ is the same to that in Corollary 9.*

In Theorems 5, as sample size $m$ increases, the term $O(e^{-m})$ decays to 0 exponentially, and the term $\Phi \left( \sqrt{m} \frac{|d^{\kappa_m^*}(\mathbb{U}, \mathbb{P}, \mathbb{Q})| - F \cdot \tau_\alpha}{\sigma_{\mathbb{U}, \mathbb{P}, \mathbb{Q}}} \right)$ converges to 1. To ensure that the test based on the learned alternative hypothesis with $F$ achieves a higher test power than that based on a uniformly selected alternative hypothesis, it is necessary to ensure a large value of $\beta > 0.5$. Fortunately, as shown in Figure 2, the probability $\beta$ can readily converge to 1 even with limited data.

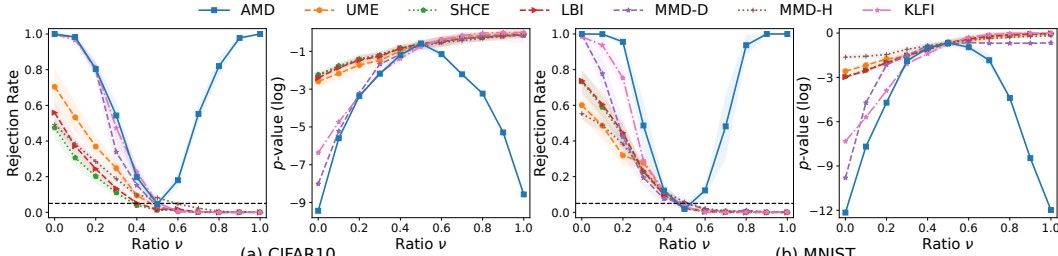

**Figure 1:** The comparisons between AMD test and baselines. We set $\mathbb{U} = \nu\mathbb{P} + (1-\nu)\mathbb{Q}$ with $\nu \in [0, 1]$. When $\nu < 0.5$, $\mathbb{Q}$ is closer to $\mathbb{U}$, and previous approaches perform well in terms of rejection rates (i.e., test power), as this aligns with the prespecified alternative hypothesis $\boldsymbol{H}_1' : d(\mathbb{U}, \mathbb{P}) > d(\mathbb{U}, \mathbb{Q})$; however, when $\nu > 0.5$, their performance deteriorates as $\mathbb{P}$ is closer to $\mathbb{U}$. In comparison, our AMD test performs well for both $\nu < 0.5$ and $\nu > 0.5$ by adjusting alternative hypothesis with $F$. Notably, when $\nu = 0.5$ (i.e., no relative similarity relationship exists), all methods control the rejection rate (type-I error) at level $\alpha = 0.05$ (black dashed line). The $p$-values align with the findings derived from the rejection rates, demonstrating the effectiveness of AMD test.

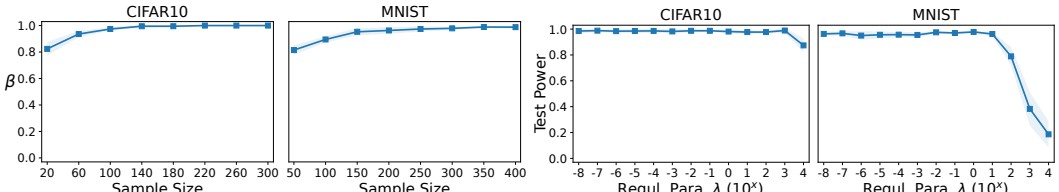

**Figure 2:** Probability $\beta = \Pr[F \cdot d^{\kappa_m^*}(\mathbb{U}, \mathbb{P}, \mathbb{Q}) > 0]$ versus sample size with parameter $\nu = 0.3$.

**Figure 3:** Influence of the regularization parameter $\lambda$ in AMD relative similarity testing.

## 5 Experiments

We first conduct experiments to compare AMD test with state-of-the-art tests on benchmark datasets, demonstrating its effectiveness. We also perform ablation studies to evaluate the contribution of each component in our method. Finally, we validate the proposed AMD test in practical applications. Notably, in all experiments, we utilize the selected deep kernels as defined in Eqn. (4). More details and results of experiments, including the results of type-I error, can be found in Appendix D.

### 5.1 Comparison with Baseline Tests on Benchmark Datasets

We begin by comparing AMD test with state-of-the-art relative similarity tests (Appendix D.1):
1) MMD-D [13, 21]; 2) KLFI [14]; 3) MMD-H [13]; 4) UME [15]; 5) SHCE [32]; 6) LBI [33].
Following the setups in [13, 14], we adapt MNIST and CIFAR10 as benchmark datasets, both of which comprise original and generative images. We set the sample size to 50 for CIFAR10 and 160 for MNIST. We denote by $\mathbb{P}$ the original images and $\mathbb{Q}$ the generative images, and set the $\mathbb{U}$ as

$$\mathbb{U} = \nu\mathbb{P} + (1-\nu)\mathbb{Q} \quad \text{with} \quad \nu \in \{0.0, 0.1, 0.2, 0.3, 0.4, 0.5, 0.6, 0.7, 0.8, 0.9, 1.0\}.$$

If $\nu < 0.5$, then the distribution $\mathbb{Q}$ is closer to $\mathbb{U}$; whereas if $\nu > 0.5$, the distribution $\mathbb{P}$ is closer to $\mathbb{U}$. When $\nu = 0.5$, $\mathbb{P}$ and $\mathbb{Q}$ are equally close to $\mathbb{U}$, and accordingly, the test should not report a significant relative similarity relationship exists (i.e., no alternative hypothesis should be accepted).

In practice, when a relative similarity relationship exists, it is unknown which distribution, $\mathbb{P}$ or $\mathbb{Q}$, is closer to $\mathbb{U}$. However, existing methods are designed to test a prespecified alternative hypothesis, e.g., $H_1' : d(\mathbb{U}, \mathbb{P}) > d(\mathbb{U}, \mathbb{Q})$. As shown in Figure 1, under this setting, they perform well in terms of rejection rate (i.e., test power) when $\nu < 0.5$, as this aligns with the assumed alternative $\boldsymbol{H}_1'$. In contrast, their performance deteriorates when $\nu > 0.5$. In comparison, our AMD test achieves high rejection rate for both $\nu < 0.5$ and $\nu > 0.5$ by adjusting the alternative hypothesis with $F$. Here, the rejection rate is larger when $|0.5 - \nu|$ is larger, since the relative similarity relationship is more pronounced and easier to detect by the test. Notably, when $\nu = 0.5$ (i.e., no relative similarity relationship exists), all methods can control the rejection rate (type-I error) at level $\alpha = 0.05$ (the black dashed line). The $p$-values in Figure 1 align with the findings derived from the test power, providing additional support for the effectiveness of AMD test.

**Table 1:** MNIST: Test power vs sample size $m$ for AMD test, with and without data augmentation, and the bold denotes the highest mean test power.

| $m$ | 130 | 160 | 190 | 220 | 250 | 280 | 310 | 340 |
|---|---|---|---|---|---|---|---|---|
| AMD | **.277** ± .043 | **.409** ± .036 | **.604** ± .049 | **.761** ± .049 | **.779** ± .048 | **.827** ± .037 | **.867** ± .028 | **.969** ± .012 |
| AMD-NA | .224 ± .038 | .353 ± .041 | .517 ± .049 | .691 ± .051 | .723 ± .060 | .790 ± .033 | .829 ± .038 | .931 ± .033 |

**Table 2:** CIFAR10: Test power vs sample size $m$ for AMD test, with and without data augmentation, and the bold denotes the highest mean test power.

| $m$ | 20 | 40 | 60 | 80 | 100 | 120 | 140 | 160 |
|---|---|---|---|---|---|---|---|---|
| AMD | **.306** ± .038 | **.477** ± .026 | **.627** ± .044 | **.775** ± .021 | **.868** ± .017 | **.871** ± .028 | **.942** ± .009 | **.968** ± .007 |
| AMD-NA | .300 ± .046 | .455 ± .028 | .590 ± .051 | .744 ± .027 | .845 ± .020 | .851 ± .021 | .919 ± .011 | .952 ± .009 |

**Ablation Studies.** Figure 2 illustrates that $\beta = \Pr[F \cdot d^{\kappa_m^*}(\mathbb{U}, \mathbb{P}, \mathbb{Q}) > 0]$ approaches 1 even with a limited data sample. This result highlights the capability of AMD test to achieve higher test power by utilizing adjusted alternative hypothesis with $F$, as supported by Theorems 5. Figure 3 illustrates the impact of the regularization parameter $\lambda$ on optimization with data augmentation. The results indicate that $\lambda$ can be selected within a relatively broad range, specifically $[10^{-8}, 10^3]$ for CIFAR10 and $[10^{-8}, 10^1]$ for MNIST. Tables 1 and 2 present comparisons of test power versus sample size for AMD test with and without augmented data (referred to as AMD-NA). The results demonstrate that incorporating augmented data facilitates improved kernel selection, enabling the test to achieve higher test power. More ablation studies, including comparisons with test on both alternative hypotheses and previous methods combined with our Phase I, are provided in Appendix D.3.

## 5.2 Performing Relative Similarity Testing in Practical Applications

We present two case studies to illustrate the practical application of our AMD test.

**Relative Model Performance Evaluation.** In the first case study (motivated by [34]), for a pre-trained ResNet50 model that performs well on the original ImageNet, we aim to assess its performance across different variants of ImageNet. A natural metric is the margin between model accuracies of the original ImageNet and its variant, with a smaller difference indicating more similar model performance on the two datasets (Appendix D.2). For the variants ImageNet-{SK, R, V2, A}, the accuracy differences, calculated using ground-truth labels, are {0.529, 0.564, 0.751, 0.827}.

Yet, obtaining ground truth labels for ImageNet variants is often challenging or costly. Given this, we illustrate that relative model performance can be evaluated using our AMD test without labels. The key is to ensure that AMD captures the same relative similarity relationships as indicated by accuracy margins, effectively supporting relative similarity testing. Here, we set the original ImageNet as $\mathbb{U}$, and sequentially set each of the variants (ImageNet-{R, V2, A, SK}) as $\mathbb{P}$. Besides, we sequentially set each of the variants (ImageNet-{SK, R, V2, A}) as $\mathbb{Q}$. By testing which distribution, $\mathbb{P}$ or $\mathbb{Q}$, is closer to $\mathbb{U}$, Figure 4 shows that our AMD achieves higher test power than MMD-D. Specifically, MMD-D fails to evaluate the relative similarity between ImageNet-SK (i.e., $\mathbb{P}$) and ImageNet-A (i.e., $\mathbb{Q}$) as it assumes $\mathbb{Q}$ is closer to $\mathbb{U}$ under the alternative hypothesis.

**Adversarial Perturbation Detection.** In the second case study, inspired by [4, 35], we illustrate that the AMD test can be used to assess the level of adversarial perturbation applied to CIFAR10. We employ ResNet18 as the base model and implement the PGD attack [36] on CIFAR-10, using perturbation levels in $\left\{\frac{i}{255}\right\}_{i=1}^{10}$. As expected, larger perturbations result in a dataset that deviates more from the original dataset. Given this, we denote the original CIFAR-10 as $\mathbb{U}$ and the 4/255-perturbed CIFAR-10 as $\mathbb{Q}$. Besides, we

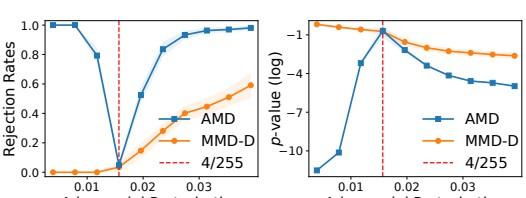

**Figure 5:** Comparisons in detecting whether the adversarial perturbations on CIFAR-10 exceeding 4/255.

set $\mathbb{P}$ as the CIFAR-10 with perturbation level in $\{i/255\}_{i=1}^{10}$ and perform testing with a sample size of 170. As shown in Figure 5, our AMD outperforms MMD-D and effectively evaluates adversarial perturbation levels. Specifically, MMD-D fails to evaluate the perturbation levels when $\mathbb{P}$ is closer to $\mathbb{U}$ (i.e., $\mathbb{P}$ is CIFAR-10 with perturbation level in $\{i/255\}_{i=1}^{3}$), as it assumes $\mathbb{Q}$ is closer to $\mathbb{U}$

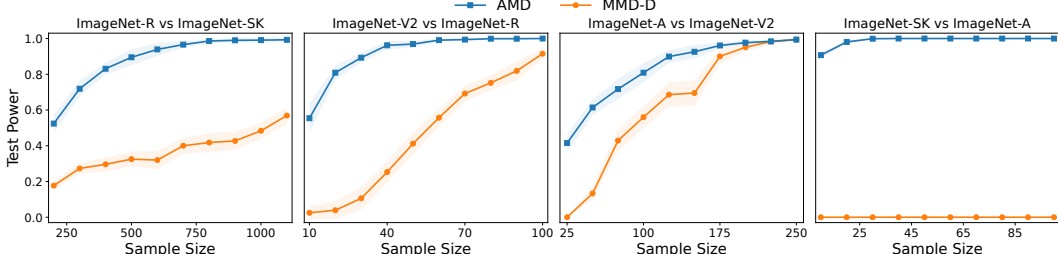

**Figure 4:** Test Power comparison between AMD and MMD-D in identifying which unlabeled variant of ImageNet the ResNet50 model (pre-trained on ImageNet) performs better on.

under the alternative hypothesis. When $\mathbb{P}$ is the CIFAR-10 with perturbation $4/255$, $\mathbb{P} = \mathbb{Q}$ and the rejection rates (type-I error) of two methods are controlled at the level $\alpha = 0.05$.

## 6    Conclusion

This work introduces a new kernel-based relative similarity test by proposing the *anchor-based maximum discrepancy* (AMD), which defines the relative similarity as the maximum discrepancy between the distances of $(\mathbb{U}, \mathbb{P})$ and $(\mathbb{U}, \mathbb{Q})$ in a space of deep kernels. Building on this metric, we learn a proper hypothesis and a kernel *simultaneously*, instead of learning a kernel after manually specifying the hypothesis. Specifically, in the AMD test, our testing procedure incorporates two phases. In Phase I, we estimate the AMD over the deep kernel space and infer the *potential hypothesis*. In Phase II, we assess the statistical significance of the potential hypothesis, where we propose a unified testing framework to derive thresholds for tests over different possible hypotheses from Phase I. We provided theoretical guarantees for the proposed method and validated its effectiveness through extensive experiments on benchmark datasets and practical applications. Looking ahead, an interesting direction for future work is to extend relative similarity testing from three distributions to the more general case of multiple distributions.

## 7    Limitation Statement

By learning a proper hypothesis and a kernel simultaneously, the AMD test could lead to overfitting, especially with limited samples. To mitigate this, we introduce augmented data in the optimization. As shown in Figure 2, the learned hypothesis reliably captures the true relative similarity even with small sample sizes (20 for CIFAR10 and 50 for MNIST). Although using augmented data increases time complexity, the experiments (Figures 4 and 5) show that only a small number of samples are needed for convergence, i.e., the test power achieves 1. In Table 13 (Appendix D), we further compare the runtime of the AMD test with all baselines and provide a detailed analysis.

In the theoretical analysis, we impose several assumptions, including the covering number assumption for Theorem 3 (Assumption 1 in Appendix A.2) and the bounded-kernel assumption for Lemma 4 and Theorem 5. These assumptions are standard and follow previous approaches [37, 13]. Although the optimization analysis in Theorem 3 reveals a gap between theoretical guarantees and practical implementation due to the inherent randomness of optimization algorithms with unknown underlying distributions, the core theoretical properties that ensure the validity of AMD in hypothesis testing do not depend on this theorem and remain robust to the outcomes of the optimization. As shown in Lemma 4, whichever deep kernel is chosen (bounded as in Eqn. (4)), type-I error is guaranteed to be controlled. Theorem 5 shows that AMD achieves higher test power whenever the learned hypothesis is better than random guessing, i.e., when $\beta > 0.5$, where the asymptotic distribution of the test samples used in the analysis is independent of the optimization on training samples [21].

## Acknowledgments and Disclosure of Funding

This research was supported by The University of Melbourne's Research Computing Services and the Petascale Campus Initiative. ZJZ and XYT are supported by the Melbourne Research Scholarship and the ARC with grant number DE240101089. LHP is supported by the ARC with grant number LP240100101. FL is supported by the ARC with grant number DE240101089, LP240100101, DP230101540 and the NSF&CSIRO Responsible AI program with grant number 2303037.

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

# Appendix

## A   Detailed Proofs for Our Theoretical Results

### A.1   Detailed Proofs of Theorem 2

*Proof.* By the definition, it is evident that $d(\mathbb{U}, \mathbb{P}, \mathbb{Q}; \mathcal{F}_{\mathcal{K}}) \geq 0$. Furthermore, since $\kappa \in \mathcal{K}$ is characteristic, the mappings

$$\mathbb{U} \to \boldsymbol{\mu}_{\mathbb{U}} = E_{\boldsymbol{z} \sim \mathbb{U}}[\kappa(\cdot, \boldsymbol{z})], \ \mathbb{P} \to \boldsymbol{\mu}_{\mathbb{P}} = E_{\boldsymbol{x} \sim \mathbb{P}}[\kappa(\cdot, \boldsymbol{x})], \quad \text{and} \quad \mathbb{Q} \to \boldsymbol{\mu}_{\mathbb{Q}} = E_{\boldsymbol{z} \sim \mathbb{Q}}[\kappa(\cdot, \boldsymbol{y})]$$

are injective. Then, for any kernel $\kappa^*$ chosen from the maximal class of kernels associated with $d(\mathbb{U}, \mathbb{P}, \mathbb{Q}; \mathcal{F}_{\mathcal{K}})$, it can be shown that

$$
\begin{aligned}
d(\mathbb{U}, \mathbb{P}, \mathbb{Q}; \mathcal{F}_{\mathcal{K}}) &= \left| \left\langle \boldsymbol{\mu}_{\mathbb{U}} - \frac{\boldsymbol{\mu}_{\mathbb{P}} + \boldsymbol{\mu}_{\mathbb{Q}}}{2}, \boldsymbol{\mu}_{\mathbb{P}} - \boldsymbol{\mu}_{\mathbb{Q}} \right\rangle_{\mathcal{H}_{\kappa^*}} \right| \\
&= \left| \left\langle \boldsymbol{\mu}_{\mathbb{U}} - \frac{\boldsymbol{\mu}_{\mathbb{Q}} + \boldsymbol{\mu}_{\mathbb{P}}}{2}, \boldsymbol{\mu}_{\mathbb{Q}} - \boldsymbol{\mu}_{\mathbb{P}} \right\rangle_{\mathcal{H}_{\kappa^*}} \right| \\
&= d(\mathbb{U}, \mathbb{Q}, \mathbb{P}; \mathcal{F}_{\mathcal{K}}) .
\end{aligned}
$$

Next, we prove that

$$d(\mathbb{U}, \mathbb{P}, \mathbb{Q}; \mathcal{F}_{\mathcal{K}}) \leq d(\mathbb{U}, \mathbb{P}, \mathbb{W}; \mathcal{F}_{\mathcal{K}}) + d(\mathbb{U}, \mathbb{Q}, \mathbb{W}; \mathcal{F}_{\mathcal{K}}) .$$

In a similar manner, denote by $\kappa'^*$ the optimal kernel chosen from the maximal class of kernels associated with $d(\mathbb{U}, \mathbb{P}, \mathbb{W}; \mathcal{F}_{\mathcal{K}})$ and let $\kappa''^*$ denote the optimal kernel chosen from the maximal class of kernels associated with $d(\mathbb{U}, \mathbb{Q}, \mathbb{W}; \mathcal{F}_{\mathcal{K}})$. The above inequality can be formalized as

$$
\left| \left\langle \boldsymbol{\mu}_{\mathbb{U}} - \frac{\boldsymbol{\mu}_{\mathbb{P}} + \boldsymbol{\mu}_{\mathbb{Q}}}{2}, \boldsymbol{\mu}_{\mathbb{P}} - \boldsymbol{\mu}_{\mathbb{Q}} \right\rangle_{\mathcal{H}_{\kappa^*}} \right|
$$
$$
\leq \left| \left\langle \boldsymbol{\mu}_{\mathbb{U}} - \frac{\boldsymbol{\mu}_{\mathbb{P}} + \boldsymbol{\mu}_{\mathbb{W}}}{2}, \boldsymbol{\mu}_{\mathbb{P}} - \boldsymbol{\mu}_{\mathbb{W}} \right\rangle_{\mathcal{H}_{\kappa'^*}} \right| + \left| \left\langle \boldsymbol{\mu}_{\mathbb{U}} - \frac{\boldsymbol{\mu}_{\mathbb{Q}} + \boldsymbol{\mu}_{\mathbb{W}}}{2}, \boldsymbol{\mu}_{\mathbb{Q}} - \boldsymbol{\mu}_{\mathbb{W}} \right\rangle_{\mathcal{H}_{\kappa''^*}} \right| .
$$

By using kernel $\kappa^*$, it is easy to see that

$$
\left| \left\langle \boldsymbol{\mu}_{\mathbb{U}} - \frac{\boldsymbol{\mu}_{\mathbb{P}} + \boldsymbol{\mu}_{\mathbb{W}}}{2}, \boldsymbol{\mu}_{\mathbb{P}} - \boldsymbol{\mu}_{\mathbb{W}} \right\rangle_{\mathcal{H}_{\kappa^*}} \right| \leq \left| \left\langle \boldsymbol{\mu}_{\mathbb{U}} - \frac{\boldsymbol{\mu}_{\mathbb{P}} + \boldsymbol{\mu}_{\mathbb{W}}}{2}, \boldsymbol{\mu}_{\mathbb{P}} - \boldsymbol{\mu}_{\mathbb{W}} \right\rangle_{\mathcal{H}_{\kappa'^*}} \right| ,
$$

and

$$
\left| \left\langle \boldsymbol{\mu}_{\mathbb{U}} - \frac{\boldsymbol{\mu}_{\mathbb{Q}} + \boldsymbol{\mu}_{\mathbb{W}}}{2}, \boldsymbol{\mu}_{\mathbb{P}} - \boldsymbol{\mu}_{\mathbb{W}} \right\rangle_{\mathcal{H}_{\kappa^*}} \right| \leq \left| \left\langle \boldsymbol{\mu}_{\mathbb{U}} - \frac{\boldsymbol{\mu}_{\mathbb{Q}} + \boldsymbol{\mu}_{\mathbb{W}}}{2}, \boldsymbol{\mu}_{\mathbb{Q}} - \boldsymbol{\mu}_{\mathbb{W}} \right\rangle_{\mathcal{H}_{\kappa''^*}} \right| .
$$

Hence, it is sufficient to prove that

$$
\left| \left\langle \boldsymbol{\mu}_{\mathbb{U}} - \frac{\boldsymbol{\mu}_{\mathbb{P}} + \boldsymbol{\mu}_{\mathbb{Q}}}{2}, \boldsymbol{\mu}_{\mathbb{P}} - \boldsymbol{\mu}_{\mathbb{Q}} \right\rangle_{\mathcal{H}_{\kappa^*}} \right|
$$
$$
\leq \left| \left\langle \boldsymbol{\mu}_{\mathbb{U}} - \frac{\boldsymbol{\mu}_{\mathbb{P}} + \boldsymbol{\mu}_{\mathbb{W}}}{2}, \boldsymbol{\mu}_{\mathbb{P}} - \boldsymbol{\mu}_{\mathbb{W}} \right\rangle_{\mathcal{H}_{\kappa^*}} \right| + \left| \left\langle \boldsymbol{\mu}_{\mathbb{U}} - \frac{\boldsymbol{\mu}_{\mathbb{Q}} + \boldsymbol{\mu}_{\mathbb{W}}}{2}, \boldsymbol{\mu}_{\mathbb{Q}} - \boldsymbol{\mu}_{\mathbb{W}} \right\rangle_{\mathcal{H}_{\kappa^*}} \right| .
$$

To facilitate further analysis, we reformulate the AMD as follows:

$$\left| \left\langle \boldsymbol{\mu}_{\mathbb{U}} - \frac{\boldsymbol{\mu}_{\mathbb{P}} + \boldsymbol{\mu}_{\mathbb{Q}}}{2}, \boldsymbol{\mu}_{\mathbb{P}} - \boldsymbol{\mu}_{\mathbb{Q}} \right\rangle_{\mathcal{H}_{\kappa^*}} \right|$$

$$= \left| \langle \boldsymbol{\mu}_{\mathbb{U}}, \boldsymbol{\mu}_{\mathbb{P}} \rangle_{\mathcal{H}_{\kappa^*}} - \langle \boldsymbol{\mu}_{\mathbb{U}}, \boldsymbol{\mu}_{\mathbb{Q}} \rangle_{\mathcal{H}_{\kappa^*}} - \frac{\|\boldsymbol{\mu}_{\mathbb{P}}\|^2_{\mathcal{H}_{\kappa^*}}}{2} + \frac{\|\boldsymbol{\mu}_{\mathbb{Q}}\|^2_{\mathcal{H}_{\kappa^*}}}{2} \right|$$

$$= \left| \frac{\|\boldsymbol{\mu}_{\mathbb{U}}\|^2_{\mathcal{H}_{\kappa^*}}}{2} - \langle \boldsymbol{\mu}_{\mathbb{U}}, \boldsymbol{\mu}_{\mathbb{Q}} \rangle_{\mathcal{H}_{\kappa^*}} + \frac{\|\boldsymbol{\mu}_{\mathbb{Q}}\|^2_{\mathcal{H}_{\kappa^*}}}{2} - \frac{\|\boldsymbol{\mu}_{\mathbb{U}}\|^2_{\mathcal{H}_{\kappa^*}}}{2} + \langle \boldsymbol{\mu}_{\mathbb{U}}, \boldsymbol{\mu}_{\mathbb{P}} \rangle_{\mathcal{H}_{\kappa^*}} - \frac{\|\boldsymbol{\mu}_{\mathbb{P}}\|^2_{\mathcal{H}_{\kappa^*}}}{2} \right|$$

$$= \left| \frac{\|\boldsymbol{\mu}_{\mathbb{U}} - \boldsymbol{\mu}_{\mathbb{Q}}\|^2_{\mathcal{H}_{\kappa^*}}}{2} - \frac{\|\boldsymbol{\mu}_{\mathbb{U}} - \boldsymbol{\mu}_{\mathbb{P}}\|^2_{\mathcal{H}_{\kappa^*}}}{2} \right| .$$

In a similar manner, we have

$$\left| \left\langle \boldsymbol{\mu}_{\mathbb{U}} - \frac{\boldsymbol{\mu}_{\mathbb{P}} + \boldsymbol{\mu}_{\mathbb{W}}}{2}, \boldsymbol{\mu}_{\mathbb{P}} - \boldsymbol{\mu}_{\mathbb{W}} \right\rangle_{\mathcal{H}_{\kappa^*}} \right| = \left| \frac{\|\boldsymbol{\mu}_{\mathbb{U}} - \boldsymbol{\mu}_{\mathbb{W}}\|^2_{\mathcal{H}_{\kappa^*}}}{2} - \frac{\|\boldsymbol{\mu}_{\mathbb{U}} - \boldsymbol{\mu}_{\mathbb{P}}\|^2_{\mathcal{H}_{\kappa^*}}}{2} \right| ,$$

and

$$\left| \left\langle \boldsymbol{\mu}_{\mathbb{U}} - \frac{\boldsymbol{\mu}_{\mathbb{Q}} + \boldsymbol{\mu}_{\mathbb{W}}}{2}, \boldsymbol{\mu}_{\mathbb{Q}} - \boldsymbol{\mu}_{\mathbb{W}} \right\rangle_{\mathcal{H}_{\kappa^*}} \right| = \left| \frac{\|\boldsymbol{\mu}_{\mathbb{U}} - \boldsymbol{\mu}_{\mathbb{W}}\|^2_{\mathcal{H}_{\kappa^*}}}{2} - \frac{\|\boldsymbol{\mu}_{\mathbb{U}} - \boldsymbol{\mu}_{\mathbb{Q}}\|^2_{\mathcal{H}_{\kappa^*}}}{2} \right| .$$

Here, without loss of generality, we assume $\|\boldsymbol{\mu}_{\mathbb{U}} - \boldsymbol{\mu}_{\mathbb{W}}\|^2_{\mathcal{H}_{\kappa^*}} \leq \|\boldsymbol{\mu}_{\mathbb{U}} - \boldsymbol{\mu}_{\mathbb{P}}\|^2_{\mathcal{H}_{\kappa^*}} \leq \|\boldsymbol{\mu}_{\mathbb{U}} - \boldsymbol{\mu}_{\mathbb{Q}}\|^2_{\mathcal{H}_{\kappa^*}}$. Then, we have

$$\left| \left\langle \boldsymbol{\mu}_{\mathbb{U}} - \frac{\boldsymbol{\mu}_{\mathbb{P}} + \boldsymbol{\mu}_{\mathbb{Q}}}{2}, \boldsymbol{\mu}_{\mathbb{P}} - \boldsymbol{\mu}_{\mathbb{Q}} \right\rangle_{\mathcal{H}_{\kappa^*}} \right| = \frac{\|\boldsymbol{\mu}_{\mathbb{U}} - \boldsymbol{\mu}_{\mathbb{Q}}\|^2_{\mathcal{H}_{\kappa^*}}}{2} - \frac{\|\boldsymbol{\mu}_{\mathbb{U}} - \boldsymbol{\mu}_{\mathbb{P}}\|^2_{\mathcal{H}_{\kappa^*}}}{2}$$

and

$$\left| \left\langle \boldsymbol{\mu}_{\mathbb{U}} - \frac{\boldsymbol{\mu}_{\mathbb{P}} + \boldsymbol{\mu}_{\mathbb{W}}}{2}, \boldsymbol{\mu}_{\mathbb{P}} - \boldsymbol{\mu}_{\mathbb{W}} \right\rangle_{\mathcal{H}_{\kappa^*}} \right| + \left| \left\langle \boldsymbol{\mu}_{\mathbb{U}} - \frac{\boldsymbol{\mu}_{\mathbb{Q}} + \boldsymbol{\mu}_{\mathbb{W}}}{2}, \boldsymbol{\mu}_{\mathbb{Q}} - \boldsymbol{\mu}_{\mathbb{W}} \right\rangle_{\mathcal{H}_{\kappa^*}} \right|$$

$$= \frac{\|\boldsymbol{\mu}_{\mathbb{U}} - \boldsymbol{\mu}_{\mathbb{Q}}\|^2_{\mathcal{H}_{\kappa^*}}}{2} + \frac{\|\boldsymbol{\mu}_{\mathbb{U}} - \boldsymbol{\mu}_{\mathbb{P}}\|^2_{\mathcal{H}_{\kappa^*}}}{2} - 2\frac{\|\boldsymbol{\mu}_{\mathbb{U}} - \boldsymbol{\mu}_{\mathbb{W}}\|^2_{\mathcal{H}_{\kappa^*}}}{2}$$

$$\geq \frac{\|\boldsymbol{\mu}_{\mathbb{U}} - \boldsymbol{\mu}_{\mathbb{Q}}\|^2_{\mathcal{H}_{\kappa^*}}}{2} - \frac{\|\boldsymbol{\mu}_{\mathbb{U}} - \boldsymbol{\mu}_{\mathbb{W}}\|^2_{\mathcal{H}_{\kappa^*}}}{2}$$

$$\geq \frac{\|\boldsymbol{\mu}_{\mathbb{U}} - \boldsymbol{\mu}_{\mathbb{Q}}\|^2_{\mathcal{H}_{\kappa^*}}}{2} - \frac{\|\boldsymbol{\mu}_{\mathbb{U}} - \boldsymbol{\mu}_{\mathbb{P}}\|^2_{\mathcal{H}_{\kappa^*}}}{2}$$

$$\geq d(\mathbb{U}, \mathbb{P}, \mathbb{Q}; \mathcal{F}_{\mathcal{K}}) .$$

For other cases where the order of the terms differs, a similar analysis applies, leading to analogous conclusions. This completes the proof. $\qquad\square$

## A.2 Detailed Proofs of Theorem 3

We start by introducing the concept of the $U$-statistic, a fundamental tool in statistics.

**Definition 6.** [37, 27] Let $h((\boldsymbol{w}_1, \boldsymbol{w}_2, \ldots, \boldsymbol{w}_r; \kappa)$ be a symmetric function of $r$ arguments. Suppose we have a random sample $\boldsymbol{w}_1, \boldsymbol{w}_2, \ldots, \boldsymbol{w}_m$ from distribution $\mathbb{W}$. The $r$-th order $U$-statistic is defined as follows

$$U_m(h) = \binom{m}{r}^{-1} \sum_{1 \leq i_1 < i_2 < \cdots < i_r \leq m} h(\boldsymbol{w}_{i_1}, \boldsymbol{w}_{i_2}, ..., \boldsymbol{w}_{i_r}) .$$

Here, $\binom{m}{r}$ is the number of ways to choose $r$ distinct indices from $m$, i.e., the binomial coefficient, and the summation is taken over all possible $r$-tuples from the sample.

Let $\mathscr{H}$ be a function space of $h(\cdot)$ for $U$-statistic and denote by pseudometric $e_m : \mathscr{H} \times \mathscr{H} \to \mathbb{R}_{\geq 0}$ as follows

$$e_m(h_1, h_2) = U_m(|h_1 - h_2|) \qquad \text{for} \qquad h_1, h_2 \in \mathscr{H} . \tag{10}$$

We define the $\varepsilon$-covering number of $(\mathscr{H}, e_m)$ as follows

$$N(\varepsilon, \mathscr{H}, e_m) = min \left\{ n : \exists h_1, h_2, ..., h_n \in \mathscr{H} \text{ s.t. } \sup_{h \in \mathscr{H}} \min_{i \leq n} e_m(h, h_i) \leq \varepsilon \right\} .$$

For $U$-statistic $U_m(h)$, we define its mean value over $\mathbb{W}$ with a fixed function $h$ as $E[U_m(h)]$, and we introduce the following theorem.

**Theorem 7.** *[37, Corollary 3.2] If $\mathscr{H}$ is a measurable class, then the conditions $E[U_m(h)] < \infty$ and $\log N(\varepsilon, \mathscr{H}, e_m)/m \to 0$ in probability imply $\|U_m(h) - E[U_m(h)]\|_{\mathscr{H}} \overset{a.s.}{\to} 0$, i.e.,*

$$\lim_{m \to \infty} \sup_{h \in \mathscr{H}} |U_m(h) - E[U_m(h)]| = 0 ,$$

*where $\overset{a.s.}{\to}$ indicates convergence almost surely.*

Recall the definition of $\widehat{d^\kappa}(Z, X, Y)$ from Eqn. (2) with corresponding samples $Z = \{z_i\}_{i=1}^m \sim \mathbb{U}^m$, $X = \{x_i\}_{i=1}^m \sim \mathbb{P}^m$ and $Y = \{y_i\}_{i=1}^m \sim \mathbb{Q}^m$; it can be equivalently expressed as a second-order $U$-statistic with $w_i = (z_i, x_i, y_i)$ and $w_j = (z_j, x_j, y_j)$ as defined in Definition 6, and is given by

$$\widehat{d^\kappa}(Z, X, Y) = \frac{1}{2m(m-1)} \sum_{i \neq j} h((z_i, x_i, y_i), (z_j, x_j, y_j); \kappa) ,$$

where we define the function

$$\begin{aligned} h((z_i, x_i, y_i), &(z_j, x_j, y_j); \kappa) \\ &= \kappa(z_i, x_j) + \kappa(z_j, x_i) - \kappa(z_i, y_j) - \kappa(z_j, y_i) - \kappa(x_i, x_j) + \kappa(y_i, y_j) , \end{aligned} \tag{11}$$

which is symmetric if the kernel $\kappa$ is symmetric.

Given the kernel space $\mathcal{K}$ consisting of bounded kernels, we denote by $\mathscr{H}$ the class of functions $h(\cdot; \kappa)$ induced by bounded kernel $\kappa \in \mathcal{K}$, as defined in Eqn. (11). Meanwhile, the corresponding pseudometric defined in Eqn. (10) can be written as

$$\begin{aligned} e_m &(h(\cdot; \kappa_1), h(\cdot; \kappa_2)) \\ &= \frac{1}{2m(m-1)} \sum_{i \neq j} |h((z_i, x_i, y_i), (z_j, x_j, y_j); \kappa_1) - h((z_i, x_i, y_i), (z_j, x_j, y_j); \kappa_2)| , \end{aligned}$$

for $h(\cdot; \kappa_1), h(\cdot; \kappa_2) \in \mathscr{H}$.

Building on this, we propose the following assumption.

**Assumption 1.** $\mathscr{H}$ is measurable and satisfies the condition $\log N(\varepsilon, \mathscr{H}, e_m)/m \overset{p}{\to} 0$.

Under the assumption, the Theorem 7 can be applied to $\widehat{d^\kappa}(Z, X, Y)$ such that

$$\lim_{m \to \infty} \sup_{h(\cdot; \kappa) \in \mathscr{H}} \left| \widehat{d^\kappa}(Z, X, Y) - d^\kappa(\mathbb{U}, \mathbb{P}, \mathbb{Q}) \right| = 0 , \tag{12}$$

where $d^\kappa(\mathbb{U}, \mathbb{P}, \mathbb{Q}) = E\left[ \widehat{d^\kappa}(Z, X, Y) \right]$

We now present the proofs of Theorem 3 under the Assumption 1 and the condition in Eqn.(12).

*Proof.* Recall that the kernel is selected as follows:

$$\kappa_m^* \in \operatorname*{arg\,max}_{\kappa \in \{\kappa_m^{*,+}, \kappa_m^{*,-}\}} \left| \widehat{d^\kappa}(Z, X, Y) \right| .$$

Here, with a regularization parameter $0 \leq \lambda < \infty$, the kernel $\kappa_m^{*,+}$ is determined by:

$$\kappa_m^{*,+} \in \operatorname*{arg\,max}_{\kappa \in \mathcal{K}} \widehat{d^\kappa}(Z, X, Y) - \lambda \cdot \widehat{d^\kappa}(Z^{\mathrm{aug}}, X^{\mathrm{aug}}, Y^{\mathrm{aug}})^2 .$$

Similarly, the kernel $\kappa_m^{*,-}$ is obtained as:

$$\kappa_m^{*,-} \in \arg\min_{\kappa \in \mathcal{K}} \widehat{d^\kappa}(Z, X, Y) + \lambda \cdot \widehat{d^\kappa}(Z^{\mathrm{aug}}, X^{\mathrm{aug}}, Y^{\mathrm{aug}})^2 .$$

Furthermore, the relative similarity relationship $F$ associated with the selected kernel is defined as follows

$$F = \begin{cases} 1 & \text{for} \quad \kappa_m^* = \kappa_m^{*,+} \\ -1 & \text{for} \quad \kappa_m^* = \kappa_m^{*,-} . \end{cases}$$

The above terms $\kappa_m^*$, $\kappa_m^{*,+}$, $\kappa_m^{*,-}$ and $F$ are all empirical estimations derived from the samples $Z$, $X$, and $Y$. To facilitate the proof, we need to define their counterparts under the true distributions $\mathbb{U}$, $\mathbb{P}$, and $\mathbb{Q}$. We start by recalling the definition of AMD in Eqn.(1) as follows:

$$
\begin{aligned}
d(\mathbb{U}, \mathbb{P}, \mathbb{Q}; \mathcal{F}_\mathcal{K}) &= \sup_{\kappa \in \mathcal{K}} \left| \left\langle \boldsymbol{\mu}_\mathbb{U} - \frac{\boldsymbol{\mu}_\mathbb{P} + \boldsymbol{\mu}_\mathbb{Q}}{2}, \boldsymbol{\mu}_\mathbb{P} - \boldsymbol{\mu}_\mathbb{Q} \right\rangle_{\mathcal{H}_\kappa} \right| \\
&= \sup_{\kappa \in \mathcal{K}} |d^\kappa(\mathbb{U}, \mathbb{P}, \mathbb{Q})| \\
&= \sup_{\kappa \in \{\kappa^{*,+}, \kappa^{*,-}\}} |d^\kappa(\mathbb{U}, \mathbb{P}, \mathbb{Q})| .
\end{aligned}
$$

In a similar manner, we define

$$\kappa^* = \arg\max_{\kappa \in \{\kappa^{*,+}, \kappa^{*,-}\}} |d^\kappa(\mathbb{U}, \mathbb{P}, \mathbb{Q})| ,$$

where

$$\kappa^{*,+} \in \arg\max_{\kappa \in \mathcal{K}} d^\kappa(\mathbb{U}, \mathbb{P}, \mathbb{Q}) \qquad \text{and} \qquad \kappa^{*,-} \in \arg\min_{\kappa \in \mathcal{K}} d^\kappa(\mathbb{U}, \mathbb{P}, \mathbb{Q}) .$$

Hence, we can write that

$$d(\mathbb{U}, \mathbb{P}, \mathbb{Q}; \mathcal{F}_\mathcal{K}) = F^* \cdot d^{\kappa^*}(\mathbb{U}, \mathbb{P}, \mathbb{Q}) ,$$

where

$$F^* = \begin{cases} 1 & \text{for} \quad \kappa^* = \kappa^{*,+} \\ -1 & \text{for} \quad \kappa^* = \kappa^{*,-} . \end{cases}$$

Building on this, to establish the convergence

$$F \cdot \widehat{d^{\kappa_m^*}}(Z, X, Y) - d(\mathbb{U}, \mathbb{P}, \mathbb{Q}; \mathcal{F}_\mathcal{K}) \xrightarrow{\text{a.s.}} 0 ,$$

it is equivalent to show that

$$F \cdot \widehat{d^{\kappa_m^*}}(Z, X, Y) - F^* \cdot d^{\kappa^*}(\mathbb{U}, \mathbb{P}, \mathbb{Q}) \xrightarrow{\text{a.s.}} 0 ,$$

which can be simplified as follows

$$
\begin{aligned}
\max\left\{ 1 \cdot \widehat{d^{\kappa_m^{*,+}}}(Z, X, Y), -1 \cdot \widehat{d^{\kappa_m^{*,-}}}(Z, X, Y) \right\} & \\
- \max\left\{ 1 \cdot d^{\kappa^{*,+}}(\mathbb{U}, \mathbb{P}, \mathbb{Q}), -1 \cdot d^{\kappa^{*,-}}(\mathbb{U}, \mathbb{P}, \mathbb{Q}) \right\} &\xrightarrow{\text{a.s.}} 0 .
\end{aligned}
$$

Here, using the *Continuous Mapping Theorem* [38] and the continuity of max function, it suffices to prove that

$$d^{\kappa_m^{*,+}}(Z, X, Y) - d^{\kappa^{*,+}}(\mathbb{U}, \mathbb{P}, \mathbb{Q}) \xrightarrow{\text{a.s.}} 0 \quad \text{and} \quad d^{\kappa_m^{*,-}}(Z, X, Y) - d^{\kappa^{*,-}}(\mathbb{U}, \mathbb{P}, \mathbb{Q}) \xrightarrow{\text{a.s.}} 0 .$$

**Building on this, we first prove that**, as $m \to \infty$, the following convergence holds:

$$\widehat{d^{\kappa_m^{*,+}}}(Z, X, Y) - d^{\kappa^{*,+}}(\mathbb{U}, \mathbb{P}, \mathbb{Q}) \xrightarrow{\text{a.s.}} 0 .$$

Based on the Borel-Cantelli Lemma [39, 40], it is sufficient to prove that

$$\sum_{m \geq 0} \Pr\left( \left| \widehat{d^{\kappa_m^{*,+}}}(Z, X, Y) - d^{\kappa^{*,+}}(\mathbb{U}, \mathbb{P}, \mathbb{Q}) \right| \geq \delta \right) < \infty , \tag{13}$$

for any $\delta > 0$.

It is evident that

$$\left| \widehat{d}^{\kappa_m^{*,+}}(Z, X, Y) - d^{\kappa^{*,+}}(\mathbb{U}, \mathbb{P}, \mathbb{Q}) \right|$$
$$\leq \left| \widehat{d}^{\kappa_m^{*,+}}(Z, X, Y) - d^{\kappa_m^{*,+}}(\mathbb{U}, \mathbb{P}, \mathbb{Q}) \right| + \left| d^{\kappa_m^{*,+}}(\mathbb{U}, \mathbb{P}, \mathbb{Q}) - d^{\kappa^{*,+}}(\mathbb{U}, \mathbb{P}, \mathbb{Q}) \right|$$
$$= A + B .$$

and we have that

$$\Pr\left( \left| \widehat{d}^{\kappa_m^{*,+}}(Z, X, Y) - d^{\kappa^{*,+}}(\mathbb{U}, \mathbb{P}, \mathbb{Q}) \right| \geq \delta \right) \leq \Pr\left( A \geq \delta/4 \right) + \Pr\left( B \geq 3\delta/4 \right) . \tag{14}$$

Based on Eqn. (12), we have that, there exists a large enough sample size $m1$ such that

$$A = \left| \widehat{d}^{\kappa_{m1}^{*,+}}(Z, X, Y) - d^{\kappa_{m1}^{*,+}}(\mathbb{U}, \mathbb{P}, \mathbb{Q}) \right| < \frac{\delta}{4} . \tag{15}$$

which indicates that

$$\Pr\left( A \geq \delta/4 \right) = 0 , \tag{16}$$

for a large enough sample size $m1$.

Notably, by the definition of $\kappa^{*,+}$, it is evident that $d^{\kappa^{*,+}}(\mathbb{U}, \mathbb{P}, \mathbb{Q}) \geq d^{\kappa_m^{*,+}}(\mathbb{U}, \mathbb{P}, \mathbb{Q})$ and we have

$$B = \left| d^{\kappa_m^{*,+}}(\mathbb{U}, \mathbb{P}, \mathbb{Q}) - d^{\kappa^{*,+}}(\mathbb{U}, \mathbb{P}, \mathbb{Q}) \right| = d^{\kappa^{*,+}}(\mathbb{U}, \mathbb{P}, \mathbb{Q}) - d^{\kappa_m^{*,+}}(\mathbb{U}, \mathbb{P}, \mathbb{Q}) .$$

Based on Eqn. (12), we have that, there exists a large enough sample size $m2$ such that

$$d^{\kappa^{*,+}}(\mathbb{U}, \mathbb{P}, \mathbb{Q}) < \widehat{d}^{\kappa^{*,+}}(Z, X, Y) + \frac{\delta}{4} . \tag{17}$$

Based on the definition of $\kappa_m^{*,+}$, we have

$$\widehat{d}^{\kappa_m^{*,+}}(Z, X, Y) - \lambda \cdot \widehat{d}^{\kappa_m^{*,+}}(Z^{\mathrm{aug}}, X^{\mathrm{aug}}, Y^{\mathrm{aug}})^2 \geq \widehat{d}^{\kappa^{*,+}}(Z, X, Y) - \lambda \cdot \widehat{d}^{\kappa^{*,+}}(Z^{\mathrm{aug}}, X^{\mathrm{aug}}, Y^{\mathrm{aug}})^2 .$$

Hence, substituting this relationship into Eqn. (17), we can refine the bound as follows

$$d^{\kappa^{*,+}}(\mathbb{U}, \mathbb{P}, \mathbb{Q})$$
$$< \widehat{d}^{\kappa_{m2}^{*,+}}(Z, X, Y) - \lambda \cdot \widehat{d}^{\kappa_{m2}^{*,+}}(Z^{\mathrm{aug}}, X^{\mathrm{aug}}, Y^{\mathrm{aug}})^2 + \lambda \cdot \widehat{d}^{\kappa^{*,+}}(Z^{\mathrm{aug}}, X^{\mathrm{aug}}, Y^{\mathrm{aug}})^2 + \frac{\delta}{4}$$
$$< \widehat{d}^{\kappa_{m2}^{*,+}}(Z, X, Y) + \lambda \cdot \widehat{d}^{\kappa^{*,+}}(Z^{\mathrm{aug}}, X^{\mathrm{aug}}, Y^{\mathrm{aug}})^2 + \frac{\delta}{4} , \tag{18}$$

for a large enough sample size $m2$.

Meanwhile, for a large enough sample size $m3$ and a constant $\lambda > 0$, the regularization term satisfies the following bound

$$\lambda \cdot \widehat{d}^{\kappa^{*,+}}(Z^{\mathrm{aug}}, X^{\mathrm{aug}}, Y^{\mathrm{aug}})^2 < \frac{\delta}{4} , \tag{19}$$

derived from Eqn. (12) and the fact that $E[\widehat{d}^{\kappa}(Z^{\mathrm{aug}}, X^{\mathrm{aug}}, Y^{\mathrm{aug}})] = 0$ for $\kappa \in \mathcal{K}$ based on Eqn. (5).

Then, combining the results of Eqns. (15), (18) and (19), the following holds

$$d^{\kappa^{*,+}}(\mathbb{U}, \mathbb{P}, \mathbb{Q}) < d^{\kappa_{m4}^{*,+}}(\mathbb{U}, \mathbb{P}, \mathbb{Q}) + \frac{\delta}{4} + \frac{\delta}{4} + \frac{\delta}{4}$$
$$= d^{\kappa_{m4}^{*,+}}(\mathbb{U}, \mathbb{P}, \mathbb{Q}) + 3\delta/4 ,$$

for a large sample size $m4 = \max\{m1, m2, m3\}$.

This indicates that

$$\Pr\left( B \geq 3\delta/4 \right) = 0 , \tag{20}$$

for a large enough sample size $m4$.

Consequently, based on Eqns. (14), (16) and (20), we have that

$$\Pr\left(\left|\widehat{d}^{\kappa^*_m,+}(Z,X,Y) - d^{\kappa^*,+}(\mathbb{U},\mathbb{P},\mathbb{Q})\right| \geq \delta\right) = 0 \ ,$$

for a large enough sample size $m4$, which implies that

$$\sum_{m \geq 0} \Pr\left(\left|\widehat{d}^{\kappa^*_m,+}(Z,X,Y) - d^{\kappa^*,+}(\mathbb{U},\mathbb{P},\mathbb{Q})\right| \geq \delta\right) < m4 < \infty \ .$$

Hence, we complete the proof of

$$\widehat{d}^{\kappa^*_m,+}(Z,X,Y) - d^{\kappa^*,+}(\mathbb{U},\mathbb{P},\mathbb{Q}) \xrightarrow{\text{a.s.}} 0 \ .$$

**In a similar manner, we now prove that**, as $m \to \infty$, the following convergence holds:

$$\widehat{d}^{\kappa^*_m,-}(Z,X,Y) - d^{\kappa^*,-}(\mathbb{U},\mathbb{P},\mathbb{Q}) \xrightarrow{\text{a.s.}} 0 \ .$$

Here, it is also sufficient to demonstrate that

$$\sum_{m \geq 0} \Pr\left(\left|\widehat{d}^{\kappa^*_m,-}(Z,X,Y) - d^{\kappa^*,-}(\mathbb{U},\mathbb{P},\mathbb{Q})\right| \geq \delta\right) < \infty \ , \tag{21}$$

for all $\delta > 0$.

It is evident that

$$\left|\widehat{d}^{\kappa^*_m,-}(Z,X,Y) - d^{\kappa^*,-}(\mathbb{U},\mathbb{P},\mathbb{Q})\right|$$
$$\leq \ \left|\widehat{d}^{\kappa^*_m,-}(Z,X,Y) - d^{\kappa^*_m,-}(\mathbb{U},\mathbb{P},\mathbb{Q})\right| + \left|d^{\kappa^*_m,-}(\mathbb{U},\mathbb{P},\mathbb{Q}) - d^{\kappa^*,-}(\mathbb{U},\mathbb{P},\mathbb{Q})\right|$$
$$= \ C + D \ .$$

and we have that

$$\Pr\left(\left|\widehat{d}^{\kappa^*_m,-}(Z,X,Y) - d^{\kappa^*,-}(\mathbb{U},\mathbb{P},\mathbb{Q})\right| \geq \delta\right) \leq \Pr\left(C \geq \delta/4\right) + \Pr\left(D \geq 3\delta/4\right) \ . \tag{22}$$

Based on Eqn. (12), we have that, there exists a large enough sample size $m5$ such that

$$C = \left|\widehat{d}^{\kappa^*_{m5},-}(Z,X,Y) - d^{\kappa^*_{m5},-}(\mathbb{U},\mathbb{P},\mathbb{Q})\right| < \frac{\delta}{4} \ . \tag{23}$$

which indicates that

$$\Pr\left(C \geq \delta/4\right) = 0 \ , \tag{24}$$

for a large enough sample size $m5$.

Notably, by the definition of $\kappa^*,-$, it is evident that $d^{\kappa^*,-}(\mathbb{U},\mathbb{P},\mathbb{Q}) \leq d^{\kappa^*_m,-}(\mathbb{U},\mathbb{P},\mathbb{Q})$ and we have

$$B = \left|d^{\kappa^*_m,-}(\mathbb{U},\mathbb{P},\mathbb{Q}) - d^{\kappa^*,-}(\mathbb{U},\mathbb{P},\mathbb{Q})\right| = d^{\kappa^*_m,-}(\mathbb{U},\mathbb{P},\mathbb{Q}) - d^{\kappa^*,-}(\mathbb{U},\mathbb{P},\mathbb{Q}) \ .$$

Based on the definition of $\kappa^*_m,-$, we have

$$\widehat{d}^{\kappa^*_m,-}(Z,X,Y) - \lambda \cdot \widehat{d}^{\kappa^*_m,-}(Z^{\text{aug}},X^{\text{aug}},Y^{\text{aug}})^2 \leq \widehat{d}^{\kappa^*,-}(Z,X,Y) - \lambda \cdot \widehat{d}^{\kappa^*,-}(Z^{\text{aug}},X^{\text{aug}},Y^{\text{aug}})^2 \ .$$

Hence, substituting this relationship into Eqn. (23), we can refine the bound as follows

$$d^{\kappa^*_{m5},-}(\mathbb{U},\mathbb{P},\mathbb{Q})$$
$$< \ \widehat{d}^{\kappa^*,-}(Z,X,Y) - \lambda \cdot \widehat{d}^{\kappa^*,-}(Z^{\text{aug}},X^{\text{aug}},Y^{\text{aug}})^2 + \lambda \cdot \widehat{d}^{\kappa^*_{m5},-}(Z^{\text{aug}},X^{\text{aug}},Y^{\text{aug}})^2 + \frac{\delta}{4}$$
$$< \ \widehat{d}^{\kappa^*,-}(Z,X,Y) + \lambda \cdot \widehat{d}^{\kappa^*_{m5},-}(Z^{\text{aug}},X^{\text{aug}},Y^{\text{aug}})^2 + \frac{\delta}{4} \ , \tag{25}$$

for a large enough sample size $m5$.

Based on Eqn. (12), we have that, there exists a large enough sample size $m6$ such that

$$d^{\kappa^*,-}(\mathbb{U},\mathbb{P},\mathbb{Q}) < \widehat{d}^{\kappa^*,-}(Z,X,Y) + \frac{\delta}{4} . \tag{26}$$

Meanwhile, for a large enough sample size $m7$ and a constant $\lambda > 0$, the regularization term satisfies the following bound

$$\lambda \cdot \widehat{d}^{\kappa^*,-}_{m7}(Z^{\mathrm{aug}},X^{\mathrm{aug}},Y^{\mathrm{aug}})^2 < \frac{\delta}{4} , \tag{27}$$

derived from Eqn. (12) and the fact that $E[\widehat{d}^{\kappa}(Z^{\mathrm{aug}},X^{\mathrm{aug}},Y^{\mathrm{aug}})] = 0$ for $\kappa \in \mathcal{K}$ based on Eqn. (5).

Then, combining the results of Eqns. (23), (25) and (27), the following holds

$$
\begin{aligned}
d^{\kappa^*,-}(\mathbb{U},\mathbb{P},\mathbb{Q}) \quad &< \quad d^{\kappa^*_{m8},-}(\mathbb{U},\mathbb{P},\mathbb{Q}) + \frac{\delta}{4} + \frac{\delta}{4} + \frac{\delta}{4} \\
&= \quad d^{\kappa^*_{m8},-}(\mathbb{U},\mathbb{P},\mathbb{Q}) + 3\delta/4 ,
\end{aligned}
$$

for a large sample size $m8 = \max\{m5, m6, m7\}$.

This indicates that

$$\Pr(D \geq 3\delta/4) = 0 , \tag{28}$$

for a large enough sample size $m8$.

Consequently, based on Eqns. (22), (24) and (28), we have that

$$\Pr\left(\left|\widehat{d}^{\kappa^*_m,-}(Z,X,Y) - d^{\kappa^*,-}(\mathbb{U},\mathbb{P},\mathbb{Q})\right| \geq \delta\right) = 0 ,$$

for a large enough sample size $m8$, which implies that

$$\sum_{m\geq0} \Pr\left(\left|\widehat{d}^{\kappa^*_m,-}(Z,X,Y) - d^{\kappa^*,-}(\mathbb{U},\mathbb{P},\mathbb{Q})\right| \geq \delta\right) < m8 < \infty .$$

Hence, we complete the proof of

$$\widehat{d}^{\kappa^*_m,-}(Z,X,Y) - d^{\kappa^*,-}(\mathbb{U},\mathbb{P},\mathbb{Q}) \xrightarrow{\mathrm{a.s.}} 0 .$$

This completes the proof of Theorem 3. $\qquad\qquad\square$

### A.3 Detailed Proofs of Lemma 4

Recall that we test on the unified hypotheses as follows

$$\boldsymbol{H}_0^{\mathrm{uni}} : F \cdot d^{\kappa^*_m}(\mathbb{U},\mathbb{P},\mathbb{Q}) \leq 0 \qquad \text{and} \qquad \boldsymbol{H}_1^{\mathrm{uni}} : F \cdot d^{\kappa^*_m}(\mathbb{U},\mathbb{P},\mathbb{Q}) > 0 .$$

We determine the testing threshold $F \cdot \tau_\alpha$ as the $(1-\alpha)$-quantile of the estimated distribution of $F \cdot \widehat{d}^{\kappa^*_m}(Z',X',Y')$ under *the proxy null hypothesis* $\boldsymbol{H}_0^{\mathrm{p}} : F \cdot d^{\kappa^*_m}(\mathbb{U},\mathbb{P},\mathbb{Q}) = 0$ (i.e., the least-favorable boundary of the composite null hypothesis) by wild bootstrap. Specifically, let $B$ be the number of bootstraps. In the $b$-th iteration ($b \in [B]$), we draw i.i.d. variables $\boldsymbol{\xi} = (\xi_1,...,\xi_m)$ from Exponential distribution with scale parameter 1 and define

$$\boldsymbol{\zeta} = \{\zeta_i\}_{i=1}^m \quad \text{with} \quad \zeta_i = m \cdot \xi_i / \sum_j^m \xi_j .$$

Then, we calculate the $b$-th wild bootstrap statistic for $\widehat{d}^{\kappa^*_m}(Z',X',Y')$ as follows

$$T_b = \sum_{i\neq j}(\zeta_i\zeta_j - 1) \times \frac{\kappa(\boldsymbol{z}'_i,\boldsymbol{x}'_j) + \kappa(\boldsymbol{z}'_j,\boldsymbol{x}_i) - \kappa(\boldsymbol{z}'_i,\boldsymbol{y}'_j) - \kappa(\boldsymbol{z}'_j,\boldsymbol{y}_i) - \kappa(\boldsymbol{x}'_i,\boldsymbol{x}_j) + \kappa(\boldsymbol{y}'_i,\boldsymbol{y}'_j)}{2m(m-1)} .$$

During such process, we obtain $B$ statistics $T_1, T_2, ..., T_B$ and introduce testing threshold $F \cdot \tau_\alpha$ with

$$\tau_\alpha = \arg\min_\tau \left\{ \sum_{b=1}^B \frac{\mathbb{I}[F \cdot T_b \leq F \cdot \tau]}{B} \geq 1 - \alpha \right\} .$$

Given the testing threshold $F \cdot \tau_\alpha > 0$, the testing procedures can be formalized as

$$\mathfrak{h}(Z', X', Y'; \kappa_m^*) = \mathbb{I}[F \cdot \widehat{d}^{\kappa_m^*}(Z', X', Y') > F \cdot \tau_\alpha] \,,$$

where $\mathfrak{h}(Z', X', Y'; \kappa_m^*) = 1$ indicates that the inferred relative similarity relationship presented by $F$ is statistically significant at level $\alpha$; otherwise, the relationship is not statistically significant.

Building on this, we introduce a useful Theorem as follows.

**Theorem 8.** *[41, Theorem 1] If $h$ is non-degenerate and $h^2(\boldsymbol{w}, \boldsymbol{w}'; \kappa) < \infty$ with $\boldsymbol{w} = (\boldsymbol{z}, \boldsymbol{x}, \boldsymbol{y})$, $\boldsymbol{w}' = (\boldsymbol{z}', \boldsymbol{x}', \boldsymbol{y}')$ and $\boldsymbol{w}, \boldsymbol{w}' \sim \mathbb{U} \times \mathbb{P} \times \mathbb{Q}$, then as $m \to \infty$, the following holds*

$$\sup_{t \in \mathbb{R}} \left| \Pr\left(\sqrt{m} F \cdot T_b \leq t\right) - \Pr\left(\sqrt{m}\left(F \cdot \widehat{d}^{\kappa_m^*}(Z', X', Y') - F \cdot d^{\kappa_m^*}(\mathbb{U}, \mathbb{P}, \mathbb{Q})\right) \leq t\right) \right| \to 0 \,.$$

We now present the proofs of Lemma 4 as follows.

*Proof.* By applying the results of Theorem 8 and substituting $t/\sqrt{m}$ with $F \cdot \tau_\alpha$, we obtain the following asymptotic behavior as $m \to \infty$:

$$\left| \Pr\left(F \cdot T_b \leq F \cdot \tau_\alpha\right) - \Pr\left(F \cdot \widehat{d}^{\kappa_m^*}(Z', X', Y') - F \cdot d^{\kappa_m^*}(\mathbb{U}, \mathbb{P}, \mathbb{Q}) \leq F \cdot \tau_\alpha\right) \right| \to 0 \,. \quad (29)$$

By the definition of $\tau_\alpha$, the following holds,

$$\Pr\left(F \cdot T_b \leq F \cdot \tau_\alpha\right) \geq 1 - \alpha \,. \quad (30)$$

Then, by applying Eqns. (29) and (30), the following holds asymptotically

$$\Pr\left(F \cdot \widehat{d}^{\kappa_m^*}(Z', X', Y') - F \cdot d^{\kappa_m^*}(\mathbb{U}, \mathbb{P}, \mathbb{Q}) \leq F \cdot \tau_\alpha\right) \geq 1 - \alpha \,,$$

which can be expressed as

$$\Pr\left(F \cdot \widehat{d}^{\kappa_m^*}(Z', X', Y') \leq F \cdot \tau_\alpha + F \cdot d^{\kappa_m^*}(\mathbb{U}, \mathbb{P}, \mathbb{Q})\right) \geq 1 - \alpha \,.$$

Under the *composite null hypothesis* $\boldsymbol{H}_0^{\mathrm{uni}} : F \cdot d^{\kappa_m^*}(\mathbb{U}, \mathbb{P}, \mathbb{Q}) \leq 0$, the ground truth $F \cdot d^{\kappa_m^*}(\mathbb{U}, \mathbb{P}, \mathbb{Q})$ is unknown, but we have that

$$\Pr\left(F \cdot \widehat{d}^{\kappa_m^*}(Z', X', Y') \leq F \cdot \tau_\alpha + F \cdot d^{\kappa_m^*}(\mathbb{U}, \mathbb{P}, \mathbb{Q})\right) \geq \Pr\left(F \cdot \widehat{d}^{\kappa_m^*}(Z', X', Y') \leq F \cdot \tau_\alpha\right) \,.$$

Hence, by considering the *proxy null hypothesis* $\boldsymbol{H}_0^{\mathrm{p}} : F \cdot d^{\kappa_m^*}(\mathbb{U}, \mathbb{P}, \mathbb{Q}) = 0$, it follows that

$$\begin{aligned}
\Pr\left(\mathfrak{h}(Z', X', Y'; \kappa_m^*) = \mathbb{I}[F \cdot \widehat{d}^{\kappa_m^*}(Z', X', Y') > F \cdot \tau_\alpha] = 1\right) \\
= \Pr\left(F \cdot \widehat{d}^{\kappa_m^*}(Z', X', Y') > F \cdot \tau_\alpha\right) \\
\leq 1 - \Pr\left(F \cdot \widehat{d}^{\kappa_m^*}(Z', X', Y') \leq F \cdot \tau_\alpha\right) \\
\leq 1 - (1 - \alpha) \\
\leq \alpha.
\end{aligned}$$

This completes the proof. $\qquad\square$

## A.4 Detailed Proofs of Theorem 5

In the proof of this theorem, we analyze a practical scenario where a relative similarity relationship is known to exist, i.e., $|d^{\kappa_m^*}(\mathbb{U}, \mathbb{P}, \mathbb{Q})| > 0$. However, the exact sign of this relationship is not known. In such cases, if we arbitrarily assume a particular direction of the relationship, for example, $d^{\kappa_m^*}(\mathbb{U}, \mathbb{P}, \mathbb{Q}) > 0$, the likelihood of this assumption being correct is 0.5.

Furthermore, in our AMD test, we denote the probability that $F$ from Phase 1 captures the true relative similarity relationship as follows

$$\beta = \Pr[F \cdot d^{\kappa_m^*}(\mathbb{U}, \mathbb{P}, \mathbb{Q}) > 0] \,.$$

Recall that, to test relative similarity, we introduce the test statistic $\widehat{d}^{\kappa_m^*}(Z', X', Y')$ (Eqn. (2)), which is an unbiased estimator of $d^{\kappa_m^*}(\mathbb{U}, \mathbb{P}, \mathbb{Q})$ and can be expressed as a $U$-statistic as follows:

$$\widehat{d}^{\kappa_m^*}(Z', X', Y') = \frac{1}{2m(m-1)} \sum_{i \neq j} h((\boldsymbol{z}_i', \boldsymbol{x}_i', \boldsymbol{y}_i'), (\boldsymbol{z}_j', \boldsymbol{x}_j', \boldsymbol{y}_j'); \kappa_m^*) \,,$$

where we define the function

$$h((\boldsymbol{z}_i', \boldsymbol{x}_i', \boldsymbol{y}_i'), (\boldsymbol{z}_j', \boldsymbol{x}_j', \boldsymbol{y}_j'); \kappa_m^*)$$
$$= \kappa_m^*(\boldsymbol{z}_i', \boldsymbol{x}_j') + \kappa_m^*(\boldsymbol{z}_j', \boldsymbol{x}_i') - \kappa_m^*(\boldsymbol{z}_i', \boldsymbol{y}_j') - \kappa_m^*(\boldsymbol{z}_j', \boldsymbol{y}_i') - \kappa_m^*(\boldsymbol{x}_i', \boldsymbol{x}_j') + \kappa_m^*(\boldsymbol{y}_i', \boldsymbol{y}_j') \,,$$

which is symmetric if the kernel $\kappa_m^*$ is symmetric.

The estimator has the following asymptotic behavior:

**Corollary 9.** *[27, Section 5] If $h$ is non-degenerate and $h^2(\boldsymbol{w}, \boldsymbol{w}'; \kappa) < \infty$ with $\boldsymbol{w} = (\boldsymbol{z}, \boldsymbol{x}, \boldsymbol{y})$, $\boldsymbol{w}' = (\boldsymbol{z}', \boldsymbol{x}', \boldsymbol{y}')$ and $\boldsymbol{w}, \boldsymbol{w}' \sim \mathbb{U} \times \mathbb{P} \times \mathbb{Q}$, the following holds*

$$\sqrt{m}\left(\widehat{d}^{\kappa_m^*}(Z', X', Y') - d^{\kappa_m^*}(\mathbb{U}, \mathbb{P}, \mathbb{Q})\right) \xrightarrow{d} \mathcal{N}(0, \sigma^2_{\mathbb{U}, \mathbb{P}, \mathbb{Q}}),$$

$$\sigma^2_{\mathbb{U}, \mathbb{P}, \mathbb{Q}} = 4\left(E_{\boldsymbol{w}}\left[E_{\boldsymbol{w}'}[h^2(\boldsymbol{w}, \boldsymbol{w}'; \kappa_m^*)]\right] - E^2_{\boldsymbol{w}, \boldsymbol{w}'}[h(\boldsymbol{w}, \boldsymbol{w}'; \kappa_m^*)]\right),$$

*where $\xrightarrow{d}$ denotes convergence in distribution.*

We now present the proofs of Theorem 5 as follows.

*Proof.* In our AMD testing procedure, as outlined in Eqn. (8), $F \cdot \tau_\alpha$ is defined as the $(1-\alpha)$-quantile of the null distribution of $F \cdot \widehat{d}^{\kappa_m^*}(Z', X', Y')$. This null distribution is estimated using the wild bootstrap under *the proxy null hypothesis $\boldsymbol{H}_0^{\mathrm{p}} : F \cdot d^{\kappa_m^*}(\mathbb{U}, \mathbb{P}, \mathbb{Q}) = 0$.*

As discussed above, in our AMD test, $\beta$ denotes the probability that $F$ from Phase 1 captures the true relative similarity relationship. Conditional on this event and based on Corollary 9, the test power of our method is given by:

$$\Pr\left(F \cdot \widehat{d}^{\kappa_m^*}(Z', X', Y') > F \cdot \tau_\alpha \,\cap\, F \cdot d^{\kappa_m^*}(\mathbb{U}, \mathbb{P}, \mathbb{Q}) > 0\right)$$
$$= \beta\Phi\left(\sqrt{m}\frac{|d^{\kappa_m^*}(\mathbb{U}, \mathbb{P}, \mathbb{Q})| - F \cdot \tau_\alpha}{\sigma_{\mathbb{U}, \mathbb{P}, \mathbb{Q}}}\right) \,.$$

Conversely, under the condition that $F$ incorrectly captures the relative similarity relationship, the test power is given by:

$$\Pr\left(F \cdot \widehat{d}^{\kappa_m^*}(Z', X', Y') > F \cdot \tau_\alpha \,\cap\, F \cdot d^{\kappa_m^*}(\mathbb{U}, \mathbb{P}, \mathbb{Q}) < 0\right)$$
$$= (1 - \beta)\Phi\left(\sqrt{m}\frac{-|d^{\kappa_m^*}(\mathbb{U}, \mathbb{P}, \mathbb{Q})| - F \cdot \tau_\alpha}{\sigma_{\mathbb{U}, \mathbb{P}, \mathbb{Q}}}\right).$$

In summary, we have the test power for our AMD test as follows

$$p = \Pr\left(F \cdot \widehat{d}^{\kappa_m^*}(Z', X', Y') > F \cdot \tau_\alpha\right)$$
$$= \beta\Phi\left(\sqrt{m}\frac{|d^{\kappa_m^*}(\mathbb{U}, \mathbb{P}, \mathbb{Q})| - F \cdot \tau_\alpha}{\sigma_{\mathbb{U}, \mathbb{P}, \mathbb{Q}}}\right) + (1 - \beta)\Phi\left(\sqrt{m}\frac{-|d^{\kappa_m^*}(\mathbb{U}, \mathbb{P}, \mathbb{Q})| - F \cdot \tau_\alpha}{\sigma_{\mathbb{U}, \mathbb{P}, \mathbb{Q}}}\right) \,.$$

For methods that manually specify an interested relative similarity relationship in alternative hypothesis (e.g., $d^{\kappa_m^*}(\mathbb{U}, \mathbb{P}, \mathbb{Q}) > 0$), it tests on

$$\boldsymbol{H}_0' : d^{\kappa_m^*}(\mathbb{U}, \mathbb{P}, \mathbb{Q}) \leq 0 \qquad \text{and} \qquad \boldsymbol{H}_1' : d^{\kappa_m^*}(\mathbb{U}, \mathbb{P}, \mathbb{Q}) > 0 \,,$$

and the likelihood of this alternative hypothesis being correct is 0.5.

To test the composite null hypothesis $\boldsymbol{H}_0' : d^{\kappa_m^*}(\mathbb{U}, \mathbb{P}, \mathbb{Q}) \leq 0$, we can set the testing threshold $\tau_\alpha' > 0$ as the $(1-\alpha)$-quantile of the asymptotic distribution of $\widehat{d}^{\kappa_m^*}(Z', X', Y')$ under the condition

$d^{\kappa_m^*}(\mathbb{U}, \mathbb{P}, \mathbb{Q}) = 0$ (i.e., the least-favorable boundary of the composite null hypothesis). Actually, this testing threshold can be replace by $F \cdot \tau_\alpha$ in practice. From Theorem 8, the testing threshold $F \cdot \tau_\alpha$ converges to the true $(1 - \alpha)$-quantile of the asymptotic distribution of $\widehat{d}^{\kappa_m^*}(Z', X', Y')$ under the proxy null hypothesis $\boldsymbol{H}_0^p : F \cdot d^{\kappa_m^*}(\mathbb{U}, \mathbb{P}, \mathbb{Q}) = 0$, which is symmetric at 0 according to Corollary 9 and matches the asymptotic distribution of $\widehat{d}^{\kappa_m^*}(Z', X', Y')$ under the condition $d^{\kappa_m^*}(\mathbb{U}, \mathbb{P}, \mathbb{Q}) = 0$. Hence, $F \cdot \tau_\alpha$ is also a consistent estimator of the $(1 - \alpha)$-quantile of the distribution of $\widehat{d}^{\kappa_m^*}(Z', X', Y')$ under the condition $d^{\kappa_m^*}(\mathbb{U}, \mathbb{P}, \mathbb{Q}) = 0$.

Given the testing threshold $F \cdot \tau_\alpha$ and conditional on the event that the specified alternative hypothesis $\boldsymbol{H}_1 : d^{\kappa_m^*}(\mathbb{U}, \mathbb{P}, \mathbb{Q}) > 0$ is correct, the test power is given by:

$$0.5\Phi\left(\sqrt{m}\frac{|d^{\kappa_m^*}(\mathbb{U}, \mathbb{P}, \mathbb{Q})| - F \cdot \tau_\alpha}{\sigma_{\mathbb{U}, \mathbb{P}, \mathbb{Q}}}\right) ,$$

from Corollary 9.

Conversely, under the condition that the specified alternative hypothesis $\boldsymbol{H}_1 : d^{\kappa_m^*}(\mathbb{U}, \mathbb{P}, \mathbb{Q}) > 0$ is false, the test power is given by:

$$0.5\Phi\left(\sqrt{m}\frac{-|d^{\kappa_m^*}(\mathbb{U}, \mathbb{P}, \mathbb{Q})| - F \cdot \tau_\alpha}{\sigma_{\mathbb{U}, \mathbb{P}, \mathbb{Q}}}\right) .$$

Hence, the test power of randomly specifying an alternative hypothesis is expressed as follows:

$$
\begin{aligned}
p' &= \Pr\left(\widehat{d}^{\kappa_m^*}(Z', X', Y') \geq F \cdot \tau_\alpha\right) \\
&= 0.5\Phi\left(\sqrt{m}\frac{|d^{\kappa_m^*}(\mathbb{U}, \mathbb{P}, \mathbb{Q})| - F \cdot \tau_\alpha}{\sigma_{\mathbb{U}, \mathbb{P}, \mathbb{Q}}}\right) + 0.5\Phi\left(\sqrt{m}\frac{-|d^{\kappa_m^*}(\mathbb{U}, \mathbb{P}, \mathbb{Q})| - F \cdot \tau_\alpha}{\sigma_{\mathbb{U}, \mathbb{P}, \mathbb{Q}}}\right) .
\end{aligned}
$$

Finally, as we can see, the difference in test power between adjusting alternative hypothesis with $F$ versus using a randomly selected alternative hypothesis is given by

$$
\begin{aligned}
p - p' &= (\beta - 0.5)\Phi\left(\sqrt{m}\frac{|d^{\kappa_m^*}(\mathbb{U}, \mathbb{P}, \mathbb{Q})| - F \cdot \tau_\alpha}{\sigma_{\mathbb{U}, \mathbb{P}, \mathbb{Q}}}\right) + (0.5 - \beta)\Phi\left(\sqrt{m}\frac{-|d^{\kappa_m^*}(\mathbb{U}, \mathbb{P}, \mathbb{Q})| - F \cdot \tau_\alpha}{\sigma_{\mathbb{U}, \mathbb{P}, \mathbb{Q}}}\right) \\
&= (\beta - 0.5)\Phi\left(\sqrt{m}\frac{|d^{\kappa_m^*}(\mathbb{U}, \mathbb{P}, \mathbb{Q})| - F \cdot \tau_\alpha}{\sigma_{\mathbb{U}, \mathbb{P}, \mathbb{Q}}}\right) + O(e^{-m}) ,
\end{aligned}
$$

where $\tau_\alpha > 0$. This completes the proof. $\qquad\square$

## A.5 Detailed Proofs of Theorem 10

In our AMD test, we denote the probability that $F$ from Phase 1 captures the true relative similarity relationship as follows

$$\beta = \Pr[F \cdot d^{\kappa_m^*}(\mathbb{U}, \mathbb{P}, \mathbb{Q}) > 0] .$$

Recall that, to test relative similarity, we introduce the test statistic $\widehat{d}^{\kappa_m^*}(Z', X', Y')$ (Eqn. (2)), which is an unbiased estimator of $d^{\kappa_m^*}(\mathbb{U}, \mathbb{P}, \mathbb{Q})$ and can be expressed as a $U$-statistic as follows:

$$\widehat{d}^{\kappa_m^*}(Z', X', Y') = \frac{1}{2m(m-1)} \sum_{i \neq j} h((\boldsymbol{z}_i', \boldsymbol{x}_i', \boldsymbol{y}_i'), (\boldsymbol{z}_j', \boldsymbol{x}_j', \boldsymbol{y}_j'); \kappa_m^*) ,$$

where we define the function

$$
\begin{aligned}
&h((\boldsymbol{z}_i', \boldsymbol{x}_i', \boldsymbol{y}_i'), (\boldsymbol{z}_j', \boldsymbol{x}_j', \boldsymbol{y}_j'); \kappa_m^*) \\
&\quad = \kappa_m^*(\boldsymbol{z}_i', \boldsymbol{x}_j') + \kappa_m^*(\boldsymbol{z}_j', \boldsymbol{x}_i') - \kappa_m^*(\boldsymbol{z}_i', \boldsymbol{y}_j') - \kappa_m^*(\boldsymbol{z}_j', \boldsymbol{y}_i') - \kappa_m^*(\boldsymbol{x}_i', \boldsymbol{x}_j') + \kappa_m^*(\boldsymbol{y}_i', \boldsymbol{y}_j') ,
\end{aligned}
$$

which is symmetric if the kernel $\kappa_m^*$ is symmetric.

The estimator has the following asymptotic behavior:

**Corollary 9'.** *[27, Section 5.5.1] If $h$ is non-degenerate and $h^2(\boldsymbol{w}, \boldsymbol{w}'; \kappa) < \infty$ with $\boldsymbol{w} = (\boldsymbol{z}, \boldsymbol{x}, \boldsymbol{y})$, $\boldsymbol{w}' = (\boldsymbol{z}', \boldsymbol{x}', \boldsymbol{y}')$ and $\boldsymbol{w}, \boldsymbol{w}' \sim \mathbb{U} \times \mathbb{P} \times \mathbb{Q}$, the following holds*

$$\sqrt{m} \left( \widehat{d}^{\kappa_m^*}(Z', X', Y') - d^{\kappa_m^*}(\mathbb{U}, \mathbb{P}, \mathbb{Q}) \right) \xrightarrow{d} \mathcal{N}(0, \sigma_{\mathbb{U}, \mathbb{P}, \mathbb{Q}}^2),$$

$$\sigma_{\mathbb{U}, \mathbb{P}, \mathbb{Q}}^2 = 4 \left( E_{\boldsymbol{w}} \left[ E_{\boldsymbol{w}'} [h^2(\boldsymbol{w}, \boldsymbol{w}'; \kappa_m^*)] \right] - E_{\boldsymbol{w}, \boldsymbol{w}'}^2 [h(\boldsymbol{w}, \boldsymbol{w}'; \kappa_m^*)] \right),$$

*where $\xrightarrow{d}$ denotes convergence in distribution.*

We now present the proofs of Theorem 10 as follows.

*Proof.* In our AMD testing procedure, as outlined in Eqn. (8), $F \cdot \tau_\alpha$ is defined as the $(1 - \alpha)$-quantile of the null distribution of $F \cdot \widehat{d}^{\kappa_m^*}(Z', X', Y')$. This null distribution is estimated using the wild bootstrap under *the proxy null hypothesis* $\boldsymbol{H}_0^{\mathrm{p}} : F \cdot d^{\kappa_m^*}(\mathbb{U}, \mathbb{P}, \mathbb{Q}) = 0$.

As discussed above, in our AMD test, $\beta$ denotes the probability that $F$ from Phase 1 captures the true relative similarity relationship. Conditional on this event and based on Corollary 9, the test power of our method is given by:

$$\Pr \left( F \cdot \widehat{d}^{\kappa_m^*}(Z', X', Y') > F \cdot \tau_\alpha \ \cap \ F \cdot d^{\kappa_m^*}(\mathbb{U}, \mathbb{P}, \mathbb{Q}) > 0 \right)$$

$$= \beta \Phi \left( \sqrt{m} \frac{|d^{\kappa_m^*}(\mathbb{U}, \mathbb{P}, \mathbb{Q})| - F \cdot \tau_\alpha}{\sigma_{\mathbb{U}, \mathbb{P}, \mathbb{Q}}} \right).$$

Conversely, under the condition that $F$ incorrectly captures the relative similarity relationship, the test power is given by:

$$\Pr \left( F \cdot \widehat{d}^{\kappa_m^*}(Z', X', Y') > F \cdot \tau_\alpha \ \cap \ F \cdot d^{\kappa_m^*}(\mathbb{U}, \mathbb{P}, \mathbb{Q}) < 0 \right)$$

$$= (1 - \beta) \Phi \left( \sqrt{m} \frac{-|d^{\kappa_m^*}(\mathbb{U}, \mathbb{P}, \mathbb{Q})| - F \cdot \tau_\alpha}{\sigma_{\mathbb{U}, \mathbb{P}, \mathbb{Q}}} \right).$$

In summary, we have the test power for our AMD test as follows

$$
\begin{aligned}
p &= \Pr \left( F \cdot \widehat{d}^{\kappa_m^*}(Z', X', Y') > F \cdot \tau_\alpha \right) \\
&= \beta \Phi \left( \sqrt{m} \frac{|d^{\kappa_m^*}(\mathbb{U}, \mathbb{P}, \mathbb{Q})| - F \cdot \tau_\alpha}{\sigma_{\mathbb{U}, \mathbb{P}, \mathbb{Q}}} \right) + (1 - \beta) \Phi \left( \sqrt{m} \frac{-|d^{\kappa_m^*}(\mathbb{U}, \mathbb{P}, \mathbb{Q})| - F \cdot \tau_\alpha}{\sigma_{\mathbb{U}, \mathbb{P}, \mathbb{Q}}} \right).
\end{aligned}
$$

For methods that test both possible alternative hypotheses in a multiple testing manner with a reduced significance level of $\alpha/2$ for each, we first test on the following hypotheses

$$\boldsymbol{H}_0' : d^{\kappa_m^*}(\mathbb{U}, \mathbb{P}, \mathbb{Q}) \le 0 \qquad \text{and} \qquad \boldsymbol{H}_1' : d^{\kappa_m^*}(\mathbb{U}, \mathbb{P}, \mathbb{Q}) > 0 \,.$$

The testing threshold $\tau_{\alpha/2}' > 0$ is determined as the $(1 - \alpha/2)$-quantile of the null distribution of $\widehat{d}^{\kappa_m^*}(Z', X', Y')$ under the condition that $d^{\kappa_m^*}(\mathbb{U}, \mathbb{P}, \mathbb{Q}) = 0$ (i.e., the least-favorable boundary of the composite null hypothesis). Given the testing threshold, we reject the null hypothesis if $\widehat{d}^{\kappa_m^*}(Z', X', Y') > \tau_{\alpha/2}'$.

Second, we also test on the following hypotheses with significance level $\alpha/2$,

$$\boldsymbol{H}_0'' : d^{\kappa_m^*}(\mathbb{U}, \mathbb{P}, \mathbb{Q}) \ge 0 \qquad \text{and} \qquad \boldsymbol{H}_1'' : d^{\kappa_m^*}(\mathbb{U}, \mathbb{P}, \mathbb{Q}) < 0 \,.$$

The testing threshold $\tau_{\alpha/2}'' < 0$ is determined as the $\alpha/2$-quantile of the null distribution of $\widehat{d}^{\kappa_m^*}(Z', X', Y')$ under the condition that $d^{\kappa_m^*}(\mathbb{U}, \mathbb{P}, \mathbb{Q}) = 0$ (i.e., the least-favorable boundary of the composite null hypothesis). Given the testing threshold, we reject the null hypothesis if $\widehat{d}^{\kappa_m^*}(Z', X', Y') < \tau_{\alpha/2}''$.

Notably, in both cases, the testing thresholds actually has the same absolute value, i.e., $\tau_{\alpha/2}' = -\tau_{\alpha/2}''$. This arises from the fact that the null distribution of $\widehat{d}^{\kappa_m^*}(Z', X', Y')$ under the condition that

$d^{\kappa_m^*}(\mathbb{U}, \mathbb{P}, \mathbb{Q}) = 0$ is symmetric at zero according to Corollary 9. Moreover, the null distribution also matches the asymptotic distribution of $F \cdot \widehat{d}^{\kappa_m^*}(Z', X', Y')$ under the condition $F \cdot d^{\kappa_m^*}(\mathbb{U}, \mathbb{P}, \mathbb{Q}) = 0$ with $F \in \{-1, +1\}$. Consequently, the absolute value of testing thresholds, i.e., $\tau'_{\alpha/2} = -\tau''_{\alpha/2}$ used in both tests can be estimated via the same wild bootstrap procedure as described in Section 3.3, with $F \in \{-1, +1\}$ and a reduced significance level of $\alpha/2$, yielding the threshold $F \cdot \tau_{\alpha/2} = \tau'_{\alpha/2} = -\tau''_{\alpha/2}$. The equivalence holds asymptotically, following from the convergence properties of $F \cdot \tau_{\alpha/2}$ established in Theorem 8.

For the two alternative hypotheses $\boldsymbol{H}'_1 : d^{\kappa_m^*}(\mathbb{U}, \mathbb{P}, \mathbb{Q}) > 0$ and $\boldsymbol{H}''_1 : d^{\kappa_m^*}(\mathbb{U}, \mathbb{P}, \mathbb{Q}) < 0$, only one of them is true, and the corresponding test exhibits the following test power from Corollary 9

$$\Phi\left(\sqrt{m}\frac{|\widehat{d}^{\kappa_m^*}(\mathbb{U}, \mathbb{P}, \mathbb{Q})| - F \cdot \tau_{\alpha/2}}{\sigma_{\mathbb{U},\mathbb{P},\mathbb{Q}}}\right).$$

Conversely, for the test with the incorrect alternative hypothesis, its test power is given by:

$$\Phi\left(\sqrt{m}\frac{-|\widehat{d}^{\kappa_m^*}(\mathbb{U}, \mathbb{P}, \mathbb{Q})| - F \cdot \tau_{\alpha/2}}{\sigma_{\mathbb{U},\mathbb{P},\mathbb{Q}}}\right).$$

Hence, the test power of testing on both possible alternative hypotheses with reduce significance level is expressed as follows

$$p'' = \Phi\left(\sqrt{m}\frac{|\widehat{d}^{\kappa_m^*}(\mathbb{U}, \mathbb{P}, \mathbb{Q})| - F \cdot \tau_{\alpha/2}}{\sigma_{\mathbb{U},\mathbb{P},\mathbb{Q}}}\right) + \Phi\left(\sqrt{m}\frac{-|\widehat{d}^{\kappa_m^*}(\mathbb{U}, \mathbb{P}, \mathbb{Q})| - F \cdot \tau_{\alpha/2}}{\sigma_{\mathbb{U},\mathbb{P},\mathbb{Q}}}\right).$$

Finally, as we can see, the difference in test power between adjusting alternative hypothesis with $F$ versus testing on both possible alternative hypotheses is given by

$$
\begin{aligned}
& p - p'' \\
= \quad & \beta\Phi\left(\sqrt{m}\frac{|\widehat{d}^{\kappa_m^*}(\mathbb{U}, \mathbb{P}, \mathbb{Q})| - F \cdot \tau_{\alpha}}{\sigma_{\mathbb{U},\mathbb{P},\mathbb{Q}}}\right) - \Phi\left(\sqrt{m}\frac{|\widehat{d}^{\kappa_m^*}(\mathbb{U}, \mathbb{P}, \mathbb{Q})| - F \cdot \tau_{\alpha/2}}{\sigma_{\mathbb{U},\mathbb{P},\mathbb{Q}}}\right) \\
& + (1-\beta)\Phi\left(\sqrt{m}\frac{-|\widehat{d}^{\kappa_m^*}(\mathbb{U}, \mathbb{P}, \mathbb{Q})| - F \cdot \tau_{\alpha}}{\sigma_{\mathbb{U},\mathbb{P},\mathbb{Q}}}\right) - \Phi\left(\sqrt{m}\frac{-|\widehat{d}^{\kappa_m^*}(\mathbb{U}, \mathbb{P}, \mathbb{Q})| - F \cdot \tau_{\alpha/2}}{\sigma_{\mathbb{U},\mathbb{P},\mathbb{Q}}}\right) \\
= \quad & \beta\Phi\left(\sqrt{m}\frac{|\widehat{d}^{\kappa_m^*}(\mathbb{U}, \mathbb{P}, \mathbb{Q})| - F \cdot \tau_{\alpha}}{\sigma_{\mathbb{U},\mathbb{P},\mathbb{Q}}}\right) - \Phi\left(\sqrt{m}\frac{|\widehat{d}^{\kappa_m^*}(\mathbb{U}, \mathbb{P}, \mathbb{Q})| - F \cdot \tau_{\alpha/2}}{\sigma_{\mathbb{U},\mathbb{P},\mathbb{Q}}}\right) + O(e^{-m}).
\end{aligned}
$$

where $F \cdot \tau_{\alpha} < F \cdot \tau_{\alpha/2}$. This completes the proof. □

# B   Relevant Work

Prior to conducting hypothesis testing, it is typically necessary to formulate the null and alternative hypotheses. Previous methods for relative similarity testing follow this procedure by *manually specifying a relationship* in the alternative hypothesis and performing the test accordingly. One approach uses a test statistic based on the difference of MMD distances: $\text{MMD}(\mathbb{U}, \mathbb{P}; \kappa) - \text{MMD}(\mathbb{U}, \mathbb{Q}; \kappa)$. If this difference exceeds a positive threshold, the test rejects the null hypothesis, essentially testing whether $\mathbb{Q}$ is closer to $\mathbb{U}$ than $\mathbb{P}$ [13]. Another approach tests on a reference distribution $\mathbb{U} = (1 - \nu)\mathbb{P} + \nu\mathbb{Q}$, where $\nu \in [0, 1]$, by evaluating whether $\nu > \delta$ for some fixed threshold $\delta \in (0, 1)$ [14]. This setup quantifies the relative closeness of $\mathbb{P}$ to the mixture distribution $\mathbb{U}$ at a level determined by $\delta$, thereby constituting an oriented test. Several subsequent methods (used as baselines in our experiments, as shown in Figure 1, with details provided in Appendix D.1), including MMD-D [21], UME [15], SHCE [32], and LBI [33], extend the framework of [13] by replacing MMD with various distance metrics from the two-sample testing literature, and likewise follow an oriented testing paradigm. In comparison, for our AMD test, we first infer the potential relative similarity relationship and then test the inferred relationship by proposing a corresponding alternative hypothesis.

By setting $\mathbb{U} = \mathbb{P}$ (or $\mathbb{U} = \mathbb{Q}$), our test can be adapted for two-sample testing, which aims to assess the difference between two distributions $\mathbb{P}$ and $\mathbb{Q}$. Some relevant approaches measures differences using

classification performance [42, 32, 43–45, 33, 46, 47], and the kernel-based approaches measure the difference between kernel embeddings of distributions [48–52].

Various techniques have been developed to select kernels for kernel-based hypothesis testing. Some of these rely on heuristic strategies, such as the median heuristic [9, 13], self-supervised representation learning [53], meta learning [54], or adaptively combine multiple kernels [11, 12]. Additionally, supervised methods have been explored, where kernels are selected based on result on held-out data [31, 3, 23]. Previous methods all selected kernels that best support the pre-specified alternative hypothesis, which poses challenges in kernel selection for relative similarity testing, as discussed in Section 1. In comparison, our AMD test learns a proper hypothesis and a kernel simultaneously, instead of learning a kernel after manually specifying the alternative hypothesis.

Wild bootstraps are widely used in hypothesis testing to approximate the null distribution [55–57]. This technique involves repeatedly re-computing the statistic with randomly assigned variables for indexes $i \in \{1, 2, ..., m\}$. Alternatively, one can use the $(1 - \alpha)$-quantile of asymptotic null distribution (e.g. Corollary 9) as the testing threshold [13, 23]. However, it is challenging to obtain an accurate asymptotic distribution with limited sample sizes.

Besides relative similarity testing, other statistical hypothesis testing methods are also widely applied in various domains of machine learning. For instance, two-sample testing could be used for out-of-distribution detection [58], adversarial image detection [35], and distribution alignment in transfer learning [22]. Independence testing could be used for domain generalization [59, 60], causal discovery [61] and trustworthy machine learning [52].

## C   Comparison with the Test on Both Possible Alternative Hypotheses

Another way to perform the relative similarity testing is to test on both possible alternative hypotheses in a multiple testing manner with reduced significance level for each test. Specifically, given the significance level $\alpha$, we can test on the following hypotheses with significance level $\alpha/2$,

$$\boldsymbol{H}'_0 : d^{\kappa^*_m}(\mathbb{U}, \mathbb{P}, \mathbb{Q}) \leq 0 \qquad \text{and} \qquad \boldsymbol{H}'_1 : d^{\kappa^*_m}(\mathbb{U}, \mathbb{P}, \mathbb{Q}) > 0 .$$

The testing threshold $\tau'_{\alpha/2} > 0$ is determined as the $(1 - \alpha/2)$-quantile of the null distribution of $\widehat{d}^{\kappa^*_m}(Z', X', Y')$ under the condition that $d^{\kappa^*_m}(\mathbb{U}, \mathbb{P}, \mathbb{Q}) = 0$ (i.e., the least-favorable boundary of the composite null hypothesis). Given the testing threshold, we reject the null hypothesis if $\widehat{d}^{\kappa^*_m}(Z', X', Y') > \tau'_{\alpha/2}$.

Second, we also test on the following hypotheses with significance level $\alpha/2$,

$$\boldsymbol{H}''_0 : d^{\kappa^*_m}(\mathbb{U}, \mathbb{P}, \mathbb{Q}) \geq 0 \qquad \text{and} \qquad \boldsymbol{H}''_1 : d^{\kappa^*_m}(\mathbb{U}, \mathbb{P}, \mathbb{Q}) < 0 .$$

The testing threshold $\tau''_{\alpha/2} < 0$ is determined as the $\alpha/2$-quantile of the null distribution of $\widehat{d}^{\kappa^*_m}(Z', X', Y')$ under the condition that $d^{\kappa^*_m}(\mathbb{U}, \mathbb{P}, \mathbb{Q}) = 0$ (i.e., the least-favorable boundary of the composite null hypothesis). Given the testing threshold, we reject the null hypothesis if $\widehat{d}^{\kappa^*_m}(Z', X', Y') < \tau''_{\alpha/2}$.

Notably, in both cases, the testing thresholds actually has the same absolute value, i.e., $\tau'_{\alpha/2} = -\tau''_{\alpha/2}$. This arises from the fact that the null distribution of $\widehat{d}^{\kappa^*_m}(Z', X', Y')$ under the condition that $d^{\kappa^*_m}(\mathbb{U}, \mathbb{P}, \mathbb{Q}) = 0$ is symmetric at zero according to Corollary 9. Moreover, the null distribution also matches the asymptotic distribution of $F \cdot \widehat{d}^{\kappa^*_m}(Z', X', Y')$ under the condition $F \cdot d^{\kappa^*_m}(\mathbb{U}, \mathbb{P}, \mathbb{Q}) = 0$ with $F \in \{-1, +1\}$. Consequently, the absolute value of testing thresholds, i.e., $\tau'_{\alpha/2} = -\tau''_{\alpha/2}$ used in both tests can be estimated via the same wild bootstrap procedure as described in Section 3.3, with $F \in \{-1, +1\}$ and a reduced significance level of $\alpha/2$, yielding the threshold $F \cdot \tau_{\alpha/2} = \tau'_{\alpha/2} = -\tau''_{\alpha/2}$. The equivalence holds asymptotically, following from the convergence properties of $F \cdot \tau_{\alpha/2}$ established in Theorem 8.

Now, we present the comparison of test power with the multiple testing procedure as follows:

**Theorem 10.** *If $d^{\kappa^*_m}(\mathbb{U}, \mathbb{P}, \mathbb{Q}) \neq 0$, the test based on the learned alternative hypothesis with $F$ achieves a higher test power than that tests on both possible alternative hypotheses is given by as*

*follows*

$$\beta\Phi\left(\sqrt{m}\frac{|d^{\kappa_m^*}(\mathbb{U},\mathbb{P},\mathbb{Q})| - F \cdot \tau_\alpha}{\sigma_{\mathbb{U},\mathbb{P},\mathbb{Q}}}\right) - \Phi\left(\sqrt{m}\frac{|d^{\kappa_m^*}(\mathbb{U},\mathbb{P},\mathbb{Q})| - F \cdot \tau_{\alpha/2}}{\sigma_{\mathbb{U},\mathbb{P},\mathbb{Q}}}\right) + O(e^{-m}),$$

*where $\sigma_{\mathbb{U},\mathbb{P},\mathbb{Q}}$ is same to that in Corollary 9 and $F \cdot \tau_\alpha < F \cdot \tau_{\alpha/2}$.*

In Theorem 10, the term $O(e^{-m})$ decays to 0 exponentially as sample size $m$ increases, and

$$\Phi\left(\sqrt{m}\frac{|d^{\kappa_m^*}(\mathbb{U},\mathbb{P},\mathbb{Q})| - F \cdot \tau_\alpha}{\sigma_{\mathbb{U},\mathbb{P},\mathbb{Q}}}\right) > \Phi\left(\sqrt{m}\frac{|d^{\kappa_m^*}(\mathbb{U},\mathbb{P},\mathbb{Q})| - F \cdot \tau_{\alpha/2}}{\sigma_{\mathbb{U},\mathbb{P},\mathbb{Q}}}\right).$$

To ensure that the test based on the learned alternative hypothesis with $F$ achieves higher test power than testing both alternative hypotheses with a reduced significance level $\alpha/2$, it is necessary to ensure a large value of $\beta > 0.5$. Fortunately, as shown in Figure 2, the probability $\beta$ can readily converge to 1 even with limited data.

## D    Additional Experimental Details and Results

### D.1    Details of State-of-the-Art Relative Similarity Tests

We compare our AMD test with state-of-the-art relative similarity tests, which include following methods:

- MMD-D: Measure relative similarity using the MMD statistic with a selected deep kernel [13];
- KLFI: Measure the relative similarity using the witness function of kernel mean embeddings of distributions, which are computed with a selected deep kernel [14].
- MMD-H: Measure relative similarity using MMD with a Gaussian kernel selected from median heuristic [13];
- UME: Evaluate the mean embeddings of distributions over test locations and measure the relative magnitudes of two distances calculated on these embeddings [15];
- SHCE: Train a binary classifier based on neural network and use a statistic about classification accuracy [32];
- LBI: Train a binary classifier based on deep neural network and use a statistic about class probabilities [33].

Several methods, including MMD-D [21], UME [15], SHCE [32], and LBI [33], are extended to relative similarity testing by following the framework of [13], replacing MMD with various distance metrics from the two-sample testing literature.

### D.2    The Definition of Accuracy Margin

We can test the accuracy margin between source dataset $S$ and target dataset $T$ for a model $f$. Let $f(x)$ represent the probability assigned by the model $f$ to the true label. We define the accuracy margin as follows

$$|E_{\boldsymbol{x} \in S}[f(\boldsymbol{x}; y_{\boldsymbol{x}})] - E_{\boldsymbol{x} \in T}[f(\boldsymbol{x}; y_{\boldsymbol{x}})]|.$$

A smaller margin indicates similar model performance in the source and target dataset.

We present the accuracy margins between the original ImageNet and its variants in Table 3, with the values computed using the pre-trained ResNet50 model.

**Table 3:** Accuracy margins between the original ImageNet and its variants.

|                 | ImageNetsk | ImageNetr | ImageNetv2 | ImageNeta |
|-----------------|------------|-----------|------------|-----------|
| Accuracy Margin | 0.529      | 0.564     | 0.751      | 0.827     |

## D.3 More Experiments

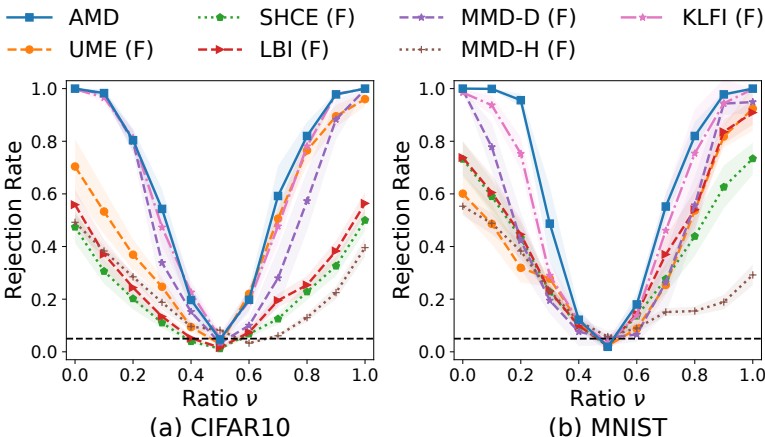

**Figure 6:** The comparisons between the AMD test and the baselines combined with the learned $F$ from Phase I of the AMD procedure (denoted by the notation $(F)$ to indicate that the hypotheses of previous baselines are now formulated based on $F$) are presented. We set $\mathbb{U} = \nu\mathbb{P} + (1 - \nu)\mathbb{Q}$ with $\nu \in [0, 1]$. When $\nu < 0.5$, $\mathbb{Q}$ is closer to $\mathbb{U}$, whereas when $\nu > 0.5$, $\mathbb{P}$ is closer to $\mathbb{U}$. All methods, including baselines augmented with $F$, perform well across both regimes $\nu < 0.5$ and $\nu > 0.5$. This contrasts with the results shown in Figure 1, where the performance of the baselines deteriorates for $\nu > 0.5$, as they are originally designed to test the prespecified alternative hypothesis $\boldsymbol{H}'_1 : d(\mathbb{U}, \mathbb{P}) > d(\mathbb{U}, \mathbb{Q})$. Notably, when $\nu = 0.5$, i.e., no relative similarity relationship exists, all methods correctly control the rejection rate (type-I error) at the nominal level $\alpha = 0.05$ (indicated by the black dashed line).

**Comparisons with baselines combined with the learned $F$ from Phase I of the AMD procedure.** We compare AMD test with baselines (Appendix D.1) combined with $F$ (denoted by the notation $(F)$ to indicate that the hypotheses of previous baselines are now formulated based on $F$) as: 1) MMD-D (F) [13, 21]; 2) KLFI (F) [14]; 3) MMD-H (F) [13]; 4) UME (F) [15]; 5) SHCE (F) [32]; 6) LBI (F) [33]. Following the setups in Figure 1, we present the rejection rates of these methods on CIFAR10 and MNIST with respect to different ratios $\nu$. In Figure 6, it is evident that our AMD test achieves higher or comparable rejection rates (i.e., test power) than others when $\nu < 0.5$ (i.e., $\mathbb{Q}$ is closer to the reference distribution $\mathbb{U}$) and $\nu > 0.5$ (i.e., $\mathbb{P}$ is closer to the reference distribution $\mathbb{U}$). Notably, when $\nu = 0.5$ (i.e., no relative similarity relationship exists and $\mathbb{P}$ and $\mathbb{Q}$ are equally close to the reference distribution $\mathbb{U}$), all methods successfully control the rejection rate (type-I error) at level $\alpha = 0.05$ (indicated by the black dashed line).

**Table 4:** CIFAR10: Test power vs ratio $\nu$ for two AMD tests: (1) with the learned alternative hypothesis using $F$ (i.e., AMD), and (2) testing both possible alternative hypotheses (denoted as AMD-B). (Part 1). The bold denotes the highest mean test power.

| $\nu$ | 0.0 | 0.1 | 0.2 | 0.3 | 0.4 |
|---|---|---|---|---|---|
| AMD | $\mathbf{1.00} \pm \mathbf{.000}$ | $\mathbf{.983} \pm \mathbf{.008}$ | $\mathbf{.804} \pm \mathbf{.022}$ | $\mathbf{.543} \pm \mathbf{.038}$ | $\mathbf{.197} \pm \mathbf{.027}$ |
| AMD-B | $\mathbf{1.00} \pm \mathbf{.000}$ | $.967 \pm .012$ | $.716 \pm .022$ | $.399 \pm .037$ | $.137 \pm .022$ |

**Table 5:** CIFAR10: Test power vs ratio $\nu$ for two AMD tests: (1) with the learned alternative hypothesis using $F$ (i.e., AMD), and (2) testing both possible alternative hypotheses (denoted as AMD-B). (Part 2). The bold denotes the highest mean test power.

| $\nu$ | 0.6 | 0.7 | 0.8 | 0.9 | 1.0 |
|---|---|---|---|---|---|
| AMD | **.180** $\pm$ **.021** | **.552** $\pm$ **.024** | **.820** $\pm$ **.027** | **.978** $\pm$ **.005** | **1.00** $\pm$ **.000** |
| AMD-B | .113 $\pm$ .013 | .427 $\pm$ .029 | .709 $\pm$ .033 | .955 $\pm$ .014 | **1.00** $\pm$ **.000** |

**Table 6:** MNIST: Test power vs ratio $\nu$ for two AMD tests: (1) with the learned alternative hypothesis using $F$ (i.e., AMD), and (2) testing both possible alternative hypotheses (denoted as AMD-B). (Part 1). The bold denotes the highest mean test power.

| $\nu$ | 0.0 | 0.1 | 0.2 | 0.3 | 0.4 |
|---|---|---|---|---|---|
| AMD | **1.00** $\pm$ **.000** | **.999** $\pm$ **.001** | **.956** $\pm$ **.012** | **.487** $\pm$ **.071** | **.122** $\pm$ **.021** |
| AMD-B | **1.00** $\pm$ **.000** | .997 $\pm$ .002 | .934 $\pm$ .016 | .379 $\pm$ .065 | .068 $\pm$ .013 |

**Table 7:** MNIST: Test power vs ratio $\nu$ for two AMD tests: (1) with the learned alternative hypothesis using $F$ (i.e., AMD), and (2) testing both possible alternative hypotheses (denoted as AMD-B). (Part 2). The bold denotes the highest mean test power.

| $\nu$ | 0.6 | 0.7 | 0.8 | 0.9 | 1.0 |
|---|---|---|---|---|---|
| AMD | **.123** $\pm$ **.016** | **.482** $\pm$ **.091** | **.937** $\pm$ **.030** | **1.00** $\pm$ **.000** | **1.00** $\pm$ **.000** |
| AMD-B | .079 $\pm$ .011 | .390 $\pm$ .080 | .904 $\pm$ .042 | .999 $\pm$ .001 | **1.00** $\pm$ **.000** |

**Comparisons with testing both possible alternative hypotheses.** Following the experimental setup in Figure 1, we compare the test power of two AMD tests: (1) with the learned alternative hypothesis using $F$ (i.e., AMD), and (2) testing both possible alternative hypotheses (denoted as AMD-B). The details of AMD-B are provided in Appendix C. From Tables 4, 5, 6 and 7, we can observe that the original AMD test consistently achieves higher or comparable test power to AMD-B across a wide range of $\nu$ values, particularly when $\nu$ is closer to 0.0 or 1.0, where the relative similarity relationship is more pronounced and $F$ is more likely to correctly identify true alternative hypothesis in Phase 1.

**Table 8:** Test power comparison of different methods in machine-generated text detection. The bold denotes the highest mean test power.

| Method | UME | SCHE | LKI | MMD-D | MMD-H | KLFI | AMD |
|---|---|---|---|---|---|---|---|
| Test Power | .791 $\pm$ .078 | .874 $\pm$ .059 | .839 $\pm$ .043 | .917 $\pm$ .036 | .869 $\pm$ .086 | .897 $\pm$ .023 | **1.00** $\pm$ **.000** |

**Comparisons in machine-generated text detection.** Following [6], we also conduct an experiment showing that our AMD performs well in detecting machine-generated text compared to other baselines. Specifically, we randomly draw sample from HC3 dataset, and assess whether the sample is machine-generated by testing its relative similarity to human-written and machine-generated texts from the TQA dataset. In this experiment, we set sample size to be 70. The test power results are shown in Table 8, where AMD test achieves higher test power compared to baselines.

**Table 9:** Test power comparison of different methods on MNIST and CIFAR10 under correct hypotheses (i.e., for baselines, the specified alternative hypothesis is correct; for AMD, the learned $F$ is correct). The bold denotes the highest mean test power.

| Dataset | MMD-D | KLFI | MMD-H | UME | SHCE | LBI | AMD |
|---------|-------|------|-------|-----|------|-----|-----|
| MNIST | .772± .112 | .897± .067 | .300± .165 | .785± .065 | .793± .082 | .710± .064 | **.968± .023** |
| CIFAR10 | .958± .076 | .936± .042 | .383± .181 | .533± .049 | .606± .069 | .671± .129 | **.983± .017** |

**Table 10:** Test power comparison of different methods on MNIST and CIFAR10 under misspecified hypotheses (i.e., for baselines, the specified alternative hypothesis is incorrect; for AMD, the learned $F$ is incorrect).

| Dataset | MMD-D | KLFI | MMD-H | UME | SHCE | LBI | AMD |
|---------|-------|------|-------|-----|------|-----|-----|
| MNIST | .006±.003 | .011±.004 | .018±.006 | .005±.002 | .047±.011 | .007±.002 | .001±.001 |
| CIFAR10 | .012±.003 | .004±.002 | .004±.002 | .017±.004 | .021±.004 | .011±.007 | .004±.002 |

**Test power comparison under correct and misspecified hypotheses.** Tables 9 and 10 compare the test power of various relative similarity testing methods on MNIST and CIFAR10 under correctly specified and misspecified hypotheses, respectively. Under the correct hypothesis setting (Table 9, i.e., for baselines, the specified alternative hypothesis is correct; for AMD, the learned $F$ is correct), AMD achieves the highest power on both datasets (0.968 on MNIST and 0.983 on CIFAR10), outperforming all baseline methods by a substantial margin. In contrast, under hypothesis misspecification (Table 10, i.e., for baselines, the specified alternative hypothesis is incorrect; for AMD, the learned $F$ is incorrect), all methods exhibit low test power and tend to accept the null hypothesis. As a result, the relative similarity relationship is not evaluated at a statistically significant level, rendering the test ineffective under such misspecification.

**Table 11:** MNIST: Test power vs. sample size $m$ for two AMD tests: (1) the optimization is divided into two separate cases: 1) $\widehat{d^\kappa}(Z, X, Y) > 0$; 2) $\widehat{d^\kappa}(Z, X, Y) < 0$ (i.e., the original AMD); (2) the squared form of test statistic is directly optimized, i.e., $\widehat{d^\kappa}(Z, X, Y)^2$, and this method is denoted as AMD-SQ. The bold denotes the highest mean test power. The bold denotes the highest mean test power.

| $m$ | 130 | 160 | 190 | 220 | 250 | 280 | 310 | 340 |
|-----|-----|-----|-----|-----|-----|-----|-----|-----|
| AMD | **.277±.043** | **.409±.036** | **.604±.049** | **.761±.049** | **.779±.048** | **.827±.037** | **.867±.028** | **.969±.012** |
| AMD-SQ | .063±.021 | .280±.078 | .392±.055 | .483±.014 | .628±.078 | .692±.048 | .787±.092 | .892±.064 |

**Table 12:** CIFAR10: Test power vs. sample size $m$ for two AMD tests: (1) the optimization is divided into two separate cases: 1) $\widehat{d^\kappa}(Z, X, Y) > 0$; 2) $\widehat{d^\kappa}(Z, X, Y) < 0$ (i.e., the original AMD); (2) the squared form of test statistic is directly optimized, i.e., $\widehat{d^\kappa}(Z, X, Y)^2$, and this method is denoted as AMD-SQ. The bold denotes the highest mean test power.

| $m$ | 20 | 40 | 60 | 80 | 100 | 120 | 140 | 160 |
|-----|-----|-----|-----|-----|-----|-----|-----|-----|
| AMD | **.306 ±.038** | **.477 ±.026** | **.627 ±.044** | **.775 ±.021** | **.868 ±.017** | **.871 ±.028** | **.942 ±.009** | **.968 ±.007** |
| AMD-SQ | .286±.054 | .452±.046 | .591±.046 | .713±.028 | .821±.011 | .834±.028 | .914±.013 | .957±.013 |

**Comparison Between Original AMD and Squared-Form Optimization (AMD-SQ).** Tables 11 and 12 present the test power of two AMD tests: (1) the optimization is divided into two separate cases: 1)$\widehat{d^\kappa}(Z, X, Y) > 0$; 2)$\widehat{d^\kappa}(Z, X, Y) < 0$ (i.e., the original AMD); (2) the squared form of the test statistic is directly optimized, i.e., $\widehat{d^\kappa}(Z, X, Y)^2$, and this method is denoted as AMD-SQ. The

test power results are presented based on MNIST and CIFAR10 with respect to different sample sizes $m$. In both datasets, the original AMD method consistently outperforms AMD-SQ. The performance gap is particularly pronounced in the low-sample regime, which is because directly optimizing the squared form of the test statistic, i.e., $\widehat{d}^\kappa(Z, X, Y)^2$, is highly sensitive to outliers or data points with large values [30], especially in high-density regions. This can overshadow important patterns in lower-density regions, potentially leading to an incomplete understanding of the data when the sample size is limited.

| Sample Size | UME (s) | SCHE (s) | MMD-D (s) | MMD-H (s) | KLFI (s) | AMD (s) |
|---|---|---|---|---|---|---|
| 50 | 792.6 | 158.7 | 93.8 | 24.3 | 104.6 | 174.3 |
| 100 | 876.2 | 214.4 | 181.1 | 31.0 | 217.9 | 411.8 |
| 150 | 931.1 | 240.9 | 224.1 | 37.8 | 274.1 | 530.7 |
| 200 | 1230.6 | 292.6 | 345.9 | 44.4 | 386.0 | 693.2 |
| 250 | 1675.0 | 313.7 | 391.5 | 50.8 | 436.0 | 766.3 |
| 300 | 2077.2 | 365.7 | 457.8 | 57.5 | 546.9 | 902.4 |

**Table 13:** Runtime comparison (in seconds) with different sample sizes on MNIST.

**Runtime Comparison.** In the Table 13, we provide a runtime comparison of AMD against baseline methods across different dataset scales on MNIST. AMD has higher time complexity than MMD-D, KLFI, and similar baselines because it includes kernel optimization on augmented data. UME is the most expensive method due to its feature selection step. In contrast, MMD-H is the fastest, as it uses a simple heuristic without any optimization. As shown in Tables 1 and 2, and Figure 4, AMD achieves high statistical power even with a relatively small sample size, significantly smaller than the full dataset. This indicates that the increased time complexity and any potential loss in power are generally acceptable in practice.

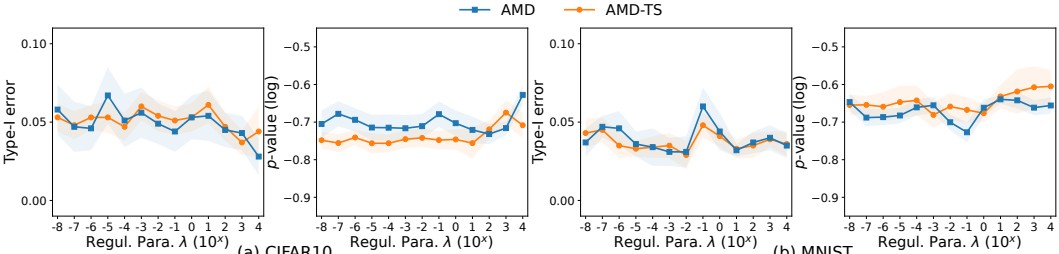

**Figure 7:** The Type-I error is controlled around $\alpha = 0.05$ w.r.t different regularization parameters for our AMD and AMD-TS tests.

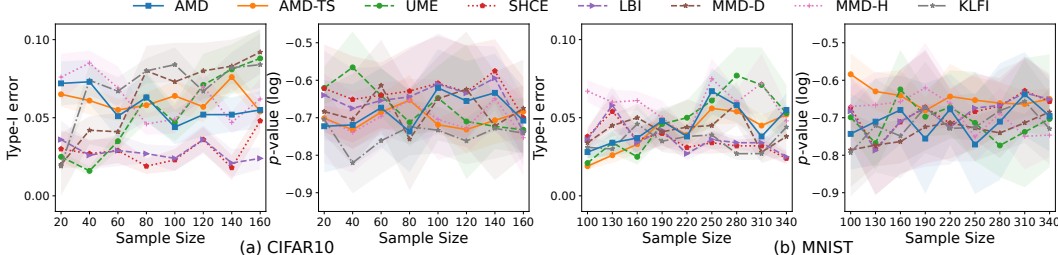

**Figure 8:** Type-I error is controlled around $\alpha = 0.05$ w.r.t different sample sizes for our AMD and the state-of-the-art relative similarity tests.

**Type-I error results.** To conduct the experiments on Type-I error, we set $\mathbb{U} = 0.5\mathbb{P} + 0.5\mathbb{Q}$, which indicates that the relative similarity relationship does not exist, i.e., $\mathbb{P}$ and $\mathbb{Q}$ are equally close to the reference distribution $\mathbb{U}$. Figure 8 demonstrates that the Type-I errors of both our AMD test and

state-of-the-art relative similarity tests are consistently controlled around $\alpha = 0.05$ across various sample sizes in relative similarity testing. Similarly, Figure 7 illustrates that the Type-I errors for our AMD and AMD-TS tests remain controlled around $\alpha = 0.05$ across different regularization parameters $\{10^{-8}, 10^{-7}, 10^{-6}, 10^{-5}, 10^{-4}, 10^{-3}, 10^{-2}, 10^{-1}, 10^{0}, 10^{1}, 10^{2}, 10^{3}, 10^{4}\}$ and datasets. These findings align well with the theoretical guarantees provided in Lemma 4. The $p$-values presented in these Figures also align with the findings derived from the Type-I error results, providing additional support for the effectiveness of our approach.

