# OpenReview forum: "Anchor-based Maximum Discrepancy for Relative Similarity Testing"
_NeurIPS.cc/2025/Conference — NeurIPS 2025 poster_

### Official Review · Reviewer_ZHeZ · 2025-06-22

**Clarity:** 3
**Significance:** 3
**Originality:** 3
**Rating:** 4
**Confidence:** 4

**Summary:**

This paper introduces a new statistical method for relative similarity testing - essentially answering "which of two datasets/distributions (P or Q) is more similar to a reference dataset (U)?" According to the author(s), existing methods have a fundamental flaw: they require one to manually specify one's hypothesis first (like "Q is closer to U than P"), then find a kernel to test it. But this approach is flawed because one can always find some kernel that will reject any hypothesis chosen, making the test meaningless. This paper addresses this issue.

**Questions:**

1. To represent a distribution, especially a complex one, what is the typical sample size?  and how are sample selected?
2. The method solves one problem (hypothesis specification bias) but potentially creates others (overfitting, computational cost, reduced power).  How do you balance these two sides?
3.The proofs are technically accurate but practically incomplete.  Do you prove properties for an idealized version that may not match the actual algorithm's behavior?

**Ethical Concerns:**

["NO or VERY MINOR ethics concerns only"]

**Final Justification:**

The submission lacked the required limitations section per the checklist. During the rebuttal phase, the authors acknowledged this omission, expressed willingness to address it, and committed to providing one.

**Limitations:**

I wish there is a summary provided.  But the authors stated that the limitations are presented throughout the paper...

**Paper Formatting Concerns:**

It would be more convenient to place your appendices behind the reference section instead of submitting a supplemental document.

**Quality:**

3

**Strengths And Weaknesses:**

Strengthens:
1. The work proposes learning both the hypothesis and the kernel simultaneously using "anchor-based maximum discrepancy (AMD)".
2. Works with samples, not full distributions.
3. Practical applications are several such as data drift detection, model comparison, and A/B testing
4. This method removes human bias from hypothesis selection, which has been a major weakness in existing approaches.

Weaknesses:
1. The method is potentially computational expensive, and sample selection can be challenging.
2. Simultaneously learning both hypothesis and kernel on the same data could lead to overfitting, especially with limited samples.
3. The paper doesn't clearly address the trade-off between avoiding manual specification bias and losing the benefits of domain expertise. Sometimes human insight about what constitutes "similarity" is actually valuable and shouldn't be discarded.
4. For the proofs documented in the appendices, the mathematical logic within stated assumptions is correct, but there are gaps between theory and practice:
--- The asymptotic theory is sound but may not apply to realistic deep kernel settings
--- Critical assumptions are never verified for the proposed method
--- The optimization/learning aspects are divorced from the statistical theory

---

> ### Author Rebuttal · Authors · 2025-07-31
>
> Many thanks for taking the time to review our manuscript. We answer your questions as follows.
>
> [Q1] The method is potentially computational expensive. The method solves one problem (hypothesis specification bias) but potentially creates others (overfitting, computational cost, reduced power). How do you balance these two sides?
>
> >[A1] Thank you for raising this question. We will include a more detailed discussion on this point in the revised version of our paper.
> >To address the fundamental challenge of kernel based relative similarity testing, namely removing human bias in hypothesis selection, we jointly learn both the hypothesis and the kernel. This added flexibility comes at a cost, as there is no free lunch. However, these costs are generally acceptable in practice. Specifically, for computational complexity, we provide a runtime comparison of AMD against baseline methods across different dataset scales using MNIST as an example.
> | Sample Size | AMD (s) | UME (s) | SCHE (s) | MMD‑D (s) | MMD‑H (s) | KLFI (s) |
> |-------|------|------|-------|--------|--------|------|
> | 50          | 174.3   | 792.6   | 158.7    | 93.8      | 24.3      | 104.6    |
> | 100         | 411.8   | 876.2   | 214.4    | 181.1     | 31.0      | 217.9    |
> | 150         | 530.7   | 931.1   | 240.9    | 224.1     | 37.8      | 274.1    |
> | 200         | 693.2   | 1230.6  | 292.6    | 345.9     | 44.4      | 386.0    |
> | 250         | 766.3   | 1675.0  | 313.7    | 391.5     | 50.8      | 436.0    |
> | 300         | 902.4   | 2077.2  | 365.7    | 457.8     | 57.5      | 546.9    |
>
> >As shown in the above Table, AMD has higher time complexity than MMD-D, KLFI, and similar baselines because it includes kernel optimization on augmented data. UME is the most expensive method due to its feature selection step. In contrast, MMD-H is the fastest, as it uses a simple heuristic without any optimization. As shown in Table 1 and Figure 4, AMD achieves high statistical power even with a relatively small sample size, significantly smaller than the full dataset. This indicates that the increased time complexity and any potential loss in power are generally acceptable in practice. In addition, regarding the concern of overfitting, Figure 2 shows that the learned hypothesis remains accurate and captures the true relative similarity relationship with high probability even when using a limited sample size (20 for CIFAR10 and 50 for MNIST). This suggests that the learned hypothesis is fairly robust.
>
> [Q2] To represent a distribution, especially a complex one, what is the typical sample size? and how are sample selected? ... sample selection can be challenging.
> >[A2] Thank you for raising this question. We find that there might be a misunderstanding regarding our paper: we do **not need to represent a distribution**. Our approach evaluates the relative similarity among three distributions: two candidates, P and Q, and a reference distribution U, by determining which of P or Q is closer to U. Instead of explicitly representing each distribution, which is challenging and requires many samples, we rely on statistical distances between distributions, making the method more sample efficient. In our setting, following standard practice, we apply the test directly to two i.i.d. samples drawn from P and Q without any specific data selection.
>
>
> [Q3] Simultaneously learning both hypothesis and kernel on the same data could lead to overfitting, especially with limited samples.
> >[A3] Thank you for your question. As demonstrated in Lemma 4, our method ensures control of the type I error. This theoretical guarantee is further supported by the empirical results presented in Figures 7 and 8 (Appendix D.4), which show that type I error remains controlled even when the sample size is small. Consequently, even if overfitting results in an incorrect alternative hypothesis being learned, the probability of incorrectly accepting this hypothesis corresponds to the type I error, which is bounded by the chosen significance level $\alpha$.
>
> [Q4] The paper doesn't clearly address the trade-off between avoiding manual specification bias and losing the benefits of domain expertise. Sometimes human insight about what constitutes "similarity" is actually valuable and shouldn't be discarded.
>
> >[A4] Thank you for your question. In this paper, our motivation is to address the problem of manual specification bias when domain knowledge is not available. In such cases, people often randomly guess an alternative hypothesis and perform the test accordingly, which can result in bias. In AMD, we learn the hypothesis, and as mentioned in [A1], Figure 2 shows that the learned hypothesis captures the true relative similarity relationship with high probability, even when using a limited sample size (20 for CIFAR10 and 50 for MNIST).
>
> >**Our method is also applicable in cases where domain knowledge is available.** In such cases, we use the manually specified hypothesis and learn the kernel directly based on it (by performing only one optimization in Phase 1: the one in line 193 or those in line 194). Then, we proceed with Phase 2 as usual. We include experimental results to demonstrate the effectiveness of AMD under this setting.
> | Dataset  | **AMD**          | MMD‑D        | KLFI         | MMD‑H        | UME          | SHCE         | LBI          |
> |----------|------------------|--------------|--------------|--------------|--------------|--------------|--------------|
> | MNIST    | **0.968 ± 0.023** | 0.772 ± 0.112 | 0.897 ± 0.067 | 0.300 ± 0.165 | 0.785 ± 0.065 | 0.793 ± 0.082 | 0.710 ± 0.064 |
> | CIFAR10  | **0.983 ± 0.017** | 0.958 ± 0.076 | 0.936 ± 0.042 | 0.383 ± 0.181 | 0.533 ± 0.049 | 0.606 ± 0.069 | 0.671 ± 0.129 |
>
> >Given the known specified hypothesis, AMD achieves the highest power on both datasets (0.968 on MNIST and 0.983 on CIFAR10).
>
> >We will revise our paper to include these clarifications and thank you again for the valuable feedback.
>
> [Q5] For the proofs documented in the appendices, the mathematical logic within stated assumptions is correct, but there are gaps between theory and practice: --- The asymptotic theory is sound but may not apply to realistic deep kernel settings --- Critical assumptions are never verified for the proposed method --- The optimization/learning aspects are divorced from the statistical theory
> >[A5] Thank you for the insightful comments. We acknowledge that narrowing the gap between theory and practice is an important direction for future work. We attempted to further investigate the behavior of the test statistic under optimization but were unsuccessful due to the challenges posed by unknown underlying distributions and the inherent randomness of optimization algorithms.
>
> >Although AMD involves an optimization procedure and assumes almost sure convergence in Theorem 3, **the core theoretical properties that ensure the validity of AMD in hypothesis testing are independent of this assumption and remain robust to the outcomes of the optimization.**
> >- **Lemma 4 (Type-I Error Control):** This result does not depend on the specific outcome of the optimization. That is, **regardless of which deep kernel (which is bounded as shown in Eqn.(5)) is selected through optimization, the type-I error is guaranteed to be controlled** (Liu et al., 2020; Huang et al., 2023). This theoretical guarantee is further supported by the empirical results presented in Figures 7 and 8 (Appendix D.4), which show that type-I error remains controlled even when the sample size is small.
> >- **Theorem 5 (Test Power Guarantee):** This theorem shows that AMD achieves higher test power **as long as the learned hypothesis is better than random guessing, i.e., when $\beta > 0.5$.** As discussed in [A1], Figure 2 empirically demonstrates that the learned hypothesis successfully captures the true relative similarity with high probability, even with a small sample size (20 for CIFAR10 and 50 for MNIST). Moreover, **the asymptotic distribution used in the test is independent of the optimization procedure and remains valid for any deep kernel obtained through optimization** (Bounliphone et al., 2016).
>
> >We will incorporate these clarifications into our revision to improve the theoretical exposition.
>
> >[1] Bounliphone W, Belilovsky E, Blaschko M, et al. A test of relative similarity for model selection in generative models. ICLR 2016.
>
> >[2] Huang B, Liu Y, Peng L. Weighted bootstrap for two-sample u-statistics. Journal of Statistical Planning and Inference, 2023.
>
> >[3] Liu F, Xu W, Lu J, et al. Learning deep kernels for non-parametric two-sample tests. ICML,  2020.
>
> [Q6] The proofs are technically accurate but practically incomplete. Do you prove properties for an idealized version that may not match the actual algorithm's behavior?
> >[A6] Thank you for this insightful question. Theoretical characterization of our algorithm’s behavior under unknown, complex distributions remains an open problem. Although we attempted such an analysis, the tight coupling between stochastic optimization dynamics and finite‑sample statistical behavior proved too challenging to resolve. We consider this an important and interesting problem, and plan to explore it in future work.
>
>
> [Q7] Limitations. I wish there is a summary provided. But the authors stated that the limitations are presented throughout the paper...
> >[A7] Thank you for the suggestion. We will add a concise summary of our paper’s limitations, including the potential problems of our method (overfitting, time complexity, and reduced power), specifically highlighting the covering‑number assumption in Theorem 3 and the bounded‑kernel requirement in Lemma 4 and Theorem 5.
>
> ---
>
> **We hope our responses have helped clarify the confusion and addressed your concerns. We would be grateful if you would kindly consider revisiting your score. If you have any further questions or need additional information, please don’t hesitate to reach out. Thank you. :)**

---

> > ### Comment · Reviewer_ZHeZ · 2025-08-03
> >
> > Dear Authors,
> >
> > I appreciate your openness about the work’s limitations. It’s clear you’re grappling with a hard and meaningful problem, so I’m updating my rating to reflect that.

---

> ### Author Response · Authors · 2025-08-03
> **Many thanks for your support!**
>
> Dear Reviewer,
>
> Thank you very much for your thoughtful and constructive feedback. We sincerely appreciate your acknowledgment of the challenges involved and your revised rating. :)

---

### Official Review · Reviewer_v8QC · 2025-07-01

**Clarity:** 1
**Significance:** 2
**Originality:** 2
**Rating:** 4
**Confidence:** 2

**Summary:**

The paper proposes anchor-based maximum discrepancy, a method for relative similarity testing that test if one distribution is closer to a reference distribution than another. To avoid the asymmetry of hypothesis formation, the paper proposes to use a new way with a constraint that allows for a symmetric formulation.

**Questions:**

- What is the rationale of the constraint in equation 1.
- What is the formula for the constraint in the kernel formulation?

**Ethical Concerns:**

["NO or VERY MINOR ethics concerns only"]

**Final Justification:**

I'd raise slightly the rating after the rebuttal. I cannot fight for its acceptance given the current state but probably it can be accepted.

**Limitations:**

yes

**Quality:**

2

**Strengths And Weaknesses:**

I find it not sufficiently clear at the rather beginning of the paper that requires clarification as follows:
- Formula (1) introduce the constraint with sgn function. I could not find the rationale of this constraint, except for the convenience of its manipulation.
- Where does the constraint disappear when used in the kernel space K?
I can not validate the paper further than this point and wait for the authors' rebuttal.

---

> ### Author Rebuttal · Authors · 2025-07-31
>
> Many thanks for taking the time to review our manuscript. We acknowledge that the **unnecessary constraint in our original presentation of formula (1) may have caused confusion**, and we appreciate your careful reading. Our intention was to highlight the connection between the AMD statistic and integral probability metrics (IPMs), but the constraint was not essential to this relationship. We address this question in detail as follows.
>
> [Q] Concerns on the formula (1)
> - Formula (1) introduce the constraint with sgn function. I could not find the rationale of this constraint, except for the convenience of its manipulation.
> - Where does the constraint disappear when used in the kernel space K? I can not validate the paper further than this point and wait for the authors' rebuttal.
> - What is the rationale of the constraint in equation 1.
> - What is the formula for the constraint in the kernel formulation?
>
> >[A] Thank you for pointing out this issue. We apologize for the confusion created by the **unnecessary constraint** in our original presentation with formula 1. Our primary intention was to illustrate the relationship between our AMD statistic (Definition 1) and the class of integral probability metrics (IPM). In fact, this relationship follows immediately from the defining equation of AMD as follows
> $$
> \sup_{f_k\in \mathcal F_{\mathcal K}}\left|\int f_k dU-\int f_kd\frac{P+Q}{2}\right|=\sup_{k\in \mathcal K}\left|\langle\mu_U-\frac{\mu_P+\mu_Q}{2},\mu_P-\mu_Q\rangle\right|\ .
> $$
> The left part of the equation can be viewed as an IPM between the distributions $U$ and $(P+Q)/2$.
>
> >For clarity, we will revise the manuscript so that we:
> >- Introduce AMD in its native form (Definition 1), without the extraneous constraint.
> >- Directly derive its equivalence to an IPM, using the equation above, rather than presenting the IPM form first and then fitting AMD into it.
>
> >We regret any confusion this may have caused. Importantly, **the methods and analyses in the paper do not rely on the removed constraint**, so this correction will not affect the correctness or validity of our main results.
>
> >Thank you again for your careful reading and for helping us improve the clarity of our presentation.
>
> ---
>
> **We hope our responses have helped clarify the confusion and addressed your concerns. We would be grateful if you would kindly consider revisiting your score. If you have any further questions or need additional information, please don’t hesitate to reach out. Thank you. :)**

---

> > ### Comment · Reviewer_v8QC · 2025-08-05
> > **Finally understood**
> >
> > I've finally got the correct formulation of the paper. I see its differences to relative similarity testing with MMD.
> > The paper still requires a revision to put it in the correct context.

---

> > > ### Author Response · Authors · 2025-08-05
> > > **Glad to hear that your concerns have been addressed**
> > >
> > > Dear Reviewer v8QC,
> > >
> > > It is very glad to know that your concerns are addressed, we will update the above content in our paper.
> > >
> > > If you have more comments or questions, just let us know. We will merge your new comments into the next version of our paper or answer new questions. It's our pleasure to have your support finally ^^
> > >
> > > Best regards,
> > >
> > > Authors of Submission 16117

---

> ### Author Response · Authors · 2025-08-05
>
> Dear Reviewer v8QC,
>
> Thank you again for your time and effort in reviewing our paper.
>
> We have carefully addressed all your comments and provided detailed responses. As the discussion period is coming to a close, we wanted to check if there are any remaining concerns preventing you from adjusting your score.
>
> If there is any additional clarification we can provide, please do not hesitate to let us know, and we will address it promptly.
>
> Best regards,
>
> Authors of Submission 16117

---

### Official Review · Reviewer_yaCd · 2025-07-03

**Clarity:** 3
**Significance:** 4
**Originality:** 4
**Rating:** 5
**Confidence:** 1

**Summary:**

This paper proposes a new relative similarity test for the question of which distribution is closer given two alternative distributions. In particular, this paper's proposed method address the existing issues in previous methods, which is ill-defined as formulating the hypothesis first and then find the correct kernel. Instead, the proposed method formulates the hypothesis and the kernel to be used at the same time. More detailly, the paper designs a two-stage method to address this issue. the effectiveness of the method is evaluated through experiments.

**Questions:**

I am not familiar with this topic, thus I am unable to find a suitable question.

**Ethical Concerns:**

["NO or VERY MINOR ethics concerns only"]

**Final Justification:**

My view keeps the same. Thanks for the paper.

**Limitations:**

not applicable.

**Quality:**

4

**Strengths And Weaknesses:**

strengths:

1. the problem this paper tries to address is very important. Given the previous problem is ill-defined, identifying the hypothesis and kernel at the same time is particular important to fully address this problem.

2. this paper has presented solid theoretical derivation to get the conclusion. The proof technique would be useful to the community.

Weakness:

1. As I am not an expert in this field, I did not find the weaknesses in this paper.

---

> ### Author Rebuttal · Authors · 2025-07-31
>
> Many thanks for your supportive comments and for taking the time to review our manuscript. :)

---

### Official Review · Reviewer_N9H2 · 2025-07-04

**Clarity:** 3
**Significance:** 2
**Originality:** 3
**Rating:** 4
**Confidence:** 3

**Summary:**

This paper introduces Anchor-based Maximum Discrepancy (AMD), a novel kernel-based metric for relative similarity testing that jointly learns both the hypothesis and the kernel rather than fixing one a priori. AMD measures the largest discrepancy between distances of two candidate distributions to an anchor distribution over a rich space of deep kernels, and its two-phase procedure first selects the kernel maximizing this discrepancy and then assesses statistical significance via a unified wild-bootstrap test. Experiments on image datasets and real-world tasks demonstrate that AMD achieves superior performance over existing baselines.

**Questions:**

1. How does the computational runtime of AMD compare to the baselines at different dataset scales?

2. See the weaknesses part.

**Ethical Concerns:**

["NO or VERY MINOR ethics concerns only"]

**Quality:**

3

**Strengths And Weaknesses:**

Strengths

1. The paper is well-constructed and easy to follow, the research objective and methodology are clearly exposed.

2. The problem studied in this paper is important, and the theoretical guarantees as well as the validation of AMD in practical applications are good.


Weaknesses

1. The computational overhead of Phase I may limit the scalability of method. The optimization procedure over a deep kernel space is likely to be significantly more expensive than baseline methods that use a fixed kernel or a simple heuristic.

2. The experiments mainly focus on image data and a small number of datasets, making it unclear how well the method generalizes to text, sequences, or more complex structured data.

---

> ### Author Rebuttal · Authors · 2025-07-31
>
> Many thanks for taking the time to review our manuscript. We  address your questions as follows.
>
> [Q1] The computational overhead of Phase I may limit the scalability of method. The optimization procedure over a deep kernel space is likely to be significantly more expensive than baseline methods that use a fixed kernel or a simple heuristic.  How does the computational runtime of AMD compare to the baselines at different dataset scales?
>
> >[A1] Thank you for raising this question. We provide a comparison of the computational runtime of AMD against the baselines at various dataset scales, using MNIST as an example. We will add the runtime analysis to our paper.
> | Sample Size | AMD (s) | UME (s) | SCHE (s) | MMD‑D (s) | MMD‑H (s) | KLFI (s) |
> |-------------|---------|---------|----------|-----------|-----------|----------|
> | 50          | 174.3   | 792.6   | 158.7    | 93.8      | 24.3      | 104.6    |
> | 100         | 411.8   | 876.2   | 214.4    | 181.1     | 31.0      | 217.9    |
> | 150         | 530.7   | 931.1   | 240.9    | 224.1     | 37.8      | 274.1    |
> | 200         | 693.2   | 1230.6  | 292.6    | 345.9     | 44.4      | 386.0    |
> | 250         | 766.3   | 1675.0  | 313.7    | 391.5     | 50.8      | 436.0    |
> | 300         | 902.4   | 2077.2  | 365.7    | 457.8     | 57.5      | 546.9    |
>
> >AMD exhibits higher time complexity compared to MMD-D, KLFI, and similar baselines, primarily due to the kernel optimization phase involving augmented data. Among all methods, UME incurs the highest computational cost, as it performs feature selection through optimization. In contrast, MMD-H demonstrates the lowest time complexity, benefiting from a heuristic kernel selection strategy that avoids any optimization.
>
> [Q2] The experiments mainly focus on image data and a small number of datasets, making it unclear how well the method generalizes to text, sequences, or more complex structured data.
>
> >[A2] Thanks for raising this question. We present an experiment over the task of machine-generated text detection. Following [Zhang et al., 2024], we randomly draw samples from the HC3 dataset and assess whether each sample is machine-generated by testing its relative similarity to human-written and machine-generated texts from the TQA dataset. In this experiment, we set the sample size to 70. The test power results are shown in the table above, where AMD achieves higher test power compared to the baselines.
> | Method        | AMD          | UME         | SCHE        | LKI         | MMD‑D       | MMD‑H       | KLFI        |
> |---------------|--------------|-------------|-------------|-------------|-------------|-------------|-------------|
> | Test Power | **1.00 ± 0.000** | 0.791 ± 0.078 | 0.874 ± 0.059 | 0.839 ± 0.043 | 0.917 ± 0.036 | 0.869 ± 0.086 | 0.897 ± 0.023 |
>
> ---
>
> **We hope our response has clarified the confusion and addressed your concerns. We would greatly appreciate it if you could kindly reconsider your score. Thank you. :)**

---

### Decision · Program_Chairs · 2025-09-17

**Decision:**

Accept (poster)

**Comment:**

The paper considers an interesting and challenging problem: testing whether P or Q is closer to U. The paper argues that existing two-sample test kind of statistics/tests are not effective. Motivated by IPMs, a new statistic (2) is proposed, which is very intuitive. Details of estimating the statistic are discussed, and provably consistent estimators are presented. Then details of a hypothesis test around this novel statistic are presented.

The paper is very well written with clear motivation, methodology and theoretical results proving the correctness. Simulations also verify the theory. Two application scenarios are also presented: Relative Model Performance Evaluation and Adversarial Perturbation Detection, highlighting the efficacy of the method.

In summary, I think the paper is a clear accept with no reservations (though some scores may appear boderline).